# EVALUATION METHODOLOGY FOR DISENTANGLED UNCERTAINTY QUANTIFICATION ON REGRESSION MODELS

## ABSTRACT

The lack of an acceptable confidence level associated with the predictions of Machine Learning (ML) models may inhibit their deployment and usage. A practical way to avoid this drawback is to enhance these predictions with trustworthiness and risk-aware add-ons such as Uncertainty Quantification (UQ). Typically, the quantified uncertainty mainly captures two intertwined parts: an epistemic uncertainty component linked to a lack of observed data and an aleatoric uncertainty component due to irreducible variability. Several existing UQ paradigms aim to disentangle the total quantified uncertainty into these two parts, with the aim of distinguishing model irrelevance from high uncertainty-level decisions. However, few of them are delving deeper into evaluating the disentanglement result, even less on real-world data. In this paper, we propose and implement a methodology to assess the effectiveness of uncertainty disentanglement through benchmarking of various UQ approaches. We introduce some indicators that allow us to robustly assess the decomposition feasibility in the absence of ground truth. The evaluation is done using an epistemic variability injection mechanism on four state-of-the-art UQ approaches based on ML models, on both synthetic and real-world gas demand datasets. The obtained results show the effectiveness of the proposed methodology for better understanding and selection of the relevant UQ approach. The corresponding code and data can be found in Github repository.

## 1 INTRODUCTION AND RELATED WORK

**Introduction** Complex systems (such as factories[16], transport[27], or electricity networks[5]) are now equipped with multiple sensors allowing subsystems characterization and global operational monitoring. To process massive amounts of dynamic data, the deployment of AI-based monitoring models becomes mandatory, raising a critical question: can we trust them? This work takes place in the context of trustworthy AI and risk-aware *Machine Learning* (ML) predictions based on *Uncertainty Quantification* (UQ). In this respect, AI-based systems must be enhanced by an uncertainty management framework[2; 12], paving the way for their certification through risk-based decision-making [9].

In the field of ML-UQ, several paradigms claim to produce models able to separate and quantify two distinct components contributing to the total uncertainty [3; 18; 10; 8]. Epistemic uncertainty expresses the irrelevance of a model facing an atypical input and aleatoric uncertainty expresses irreducible intrinsic variability in a model decision [15]. These two components are defined *for and in* a given modeling scope, resulting from the methodological choice of both observed features and model type, which depends on the predictive task's characteristics. Setting up this modeling scope draws the line between upstream sources of irreducible uncertainty (e.g. stemming from the studied phenomena, measurement imprecision, unobserved hidden variables, or limiting model hypothesis) and reducible sources of uncer-

tainty (e.g. out-of-distribution data or lack of observations). According to a common view in the ML-UQ community [15; 18; 8], *disentangled Uncertainty Quantification* (dUQ) approaches are often composed of two parts. The first is an explicit or implicit ensemble of submodels, each one providing a prediction and an aleatoric estimation. The second is a metamodel synthesizing the ensemble outputs and producing an epistemic confidence level based on the variability of ensemble decisions. To highlight the meaning of this distinction, we asked the following question: "for a given modeling scope, can we reduce the variability of predictions by enhancing the quantity and/or quality of observed data?"

However, dUQ faces both technical and methodological difficulties. On real data with noisy and limited observations, the epistemic and aleatoric uncertainty components are strongly entangled [15; 9]. Plus, there is no ground truth that allows the evaluation of the quantification and even less the possibility of the decomposition. In this regard, our contribution addresses the following methodological challenges, related to robust evaluation of model epistemic confidence, despite the absence of ground truth in real data. We will first compare recent works on UQ regression in ML, then propose a dUQ evaluation methodology based on a novel epistemic variability injection mechanism, aiming to exhibit or disprove the effectiveness of aleatoric and epistemic uncertainty disentanglement.

**Related Works**   Multiple survey papers are also dedicated to UQ in ML [2; 12] and focus on three main UQ paradigms. The *Bayesian* formalism [6; 28] is widely used to develop probabilistic methods for UQ. As the exact Bayesian inference is intractable, multiple approximations are proposed in literature. *Monte Carlo Dropout* (MCDP) is one of recent attempts to estimate the uncertainty of forecast using Dropout in neural networks [11; 28]. The *ensemble models* [17] are widely used for uncertainty estimation due to their simple implementation. The uncertainty could be measured through the prediction confidence of the ensemble members. The well-known ensemble approach, namely, *Random Forests* (RF) can be used for the estimation of uncertainty indicators based on the total variance theorem [19; 26]. Finally, *Evidential Deep Learning* (EDL) [25] learns a distribution over the parametric space of model outcomes and collects evidence regarding the model predictions. Recently, the evidential formalism has been adapted for regression problems in [3] (see section B of Appendix).

Then, we propose to summarize the recent literature through three main categories of characteristics: features, problem support, and environment configuration (see Table 1). Each of these categories has its own set of criteria, allowing a better analysis of each approach. The last line has been added to illustrate our contribution through the proposed dUQ evaluation, and the benchmark carried out on it.

Table 1: Summary of UQ approaches and the characteristics covered by proposed benchmark framework.

| | Methods | Features | | | Problem support | | Environment setup | | | |
|---|---|---|---|---|---|---|---|---|---|---|
| | | UQ paradigm | Prior | UQ Decomposition | Regression | Classification | Dataset | Evaluation criteria | Interpretation | Baselines |
| **UQ Papers** | MCDP [11] | Drop-out | No | No | Yes | Yes | Public / Synthetic (diverse) | NLL / RMSE | Local | Yes (diverse) |
| | BNN+LV [8] | Bayesian | Yes | Yes | Yes | No | Public / Synthetic (diverse) | NLL | Local | Variants |
| | DeepSTUQ [23] | Variational Ensemble | Yes | Yes | Yes | No | Public | NLL/Coverage Sharpness | Partial Quantitative | Yes (diverse) |
| | DeepEnsemble [17] | Ensemble | No | No | Yes | Yes | Public / Synthetic (stationary) | NLL / RMSE Brier/Calibration | Local Qualitative | Yes (diverse) |
| | AutoDEUQ [10] | Ensemble | No | Yes | Yes | No | Public / Synthetic (stationary) | NLL/RMSE | No | Yes (diverse) |
| | PI [21] | Ensemble | No | No | Yes | No | Public / Synthetic (stationary) | Sharpness Coverage | No | Variants |
| | Kalman [24] | Covariance | Yes | Yes | Yes | No | Multivariate (Time series) | Relative Error | No | Variants |
| | EDL [3] | Evidential | Yes | No | Yes | No | Public / Synthetic (stationary) | NLL/RMSE | Partial Quantitative | Yes (diverse) |
| **Ours** | ∅ | Benchmark Framework | - | Yes | Yes | No | Real Public / Synthetic (Time series) | Sharpness Coverage NLL/RMSE | Cross-comparison Qualitative Quantitative | Yes (diverse) |

Color codes (green: satisfying, orange: partial, red: ignorance)

From Table 1, we can observe the existence of a diverse range of paradigms (from Drop-out to Ensemble and Evidential), each having its own characteristics. Some consider the distinction between two sources

of uncertainty in their modeling, whereas others output a unique quantity presenting the total uncertainty. The regression problem is mostly addressed, whereas the adaptation to classification is straightforward. Furthermore, most approaches are evaluated on synthetic or small public datasets with some limitations in the evaluation methodology. Firstly, those datasets do not reflect the underlying complexity of real-world data due to inherent noises from heterogeneous sources (human, inherent, missing knowledge, etc.). Additionally, there is a lack of interpretation and behavioral analysis of uncertainty components during the evaluation (in particular regarding the epistemic part), which often focuses on a global *Negative Log Likelihood* (NLL) metric. The influence and appropriateness of evaluation metrics for UQ models are also studied in the literature [13; 4; 20], based on the use of reliability diagrams (or reliability based on the local fit and density principles) and calibrated measures for comparison of various approaches. However, these uncertainty metrics are not compared through various UQ paradigms.

In this article, we seek to bridge this gap by introducing a dUQ evaluation methodology providing a comprehensive benchmark of various typologies of UQ models. To this end, we apply our methodology on four standard ML approaches which are *Monte Carlo Dropout* (MCDP), Deep Ensemble, *Random Forest disentangled Uncertainty Quantification* (RF-dUQ), EDL, and using real and synthetic datasets.

## 2 DISENTANGLED UNCERTAINY QUANTIFICATION MODELING FRAMEWORK

Before presenting how dUQ approaches work, we provide technical details concerning uncertainty sources in the context of a regression task, and how a model captures these as aleatoric or epistemic. Let us consider a modeling framework, in which random variables (denoted $\varepsilon$) are linked to specific uncertainty sources. Here, time series forecasting (link to the type of data on which the work was undertaken) is treated as a regression problem based on time-dependent features. In this context, a model $\hat{f}$ aims to predict the nominal behavior for variables of interest represented by univariate/multivariate time series $Y = (y_1, \ldots, y_t, \ldots, y_T)$. The forecast at time step $t$ for the variable $y_t$ will be based on a vector of observed variables $\mathbf{x}_t$ composed of both exogenous $\mathbf{c}_t$ and lagged response $Y_t^{past} = (y_{t-lag}, \ldots, y_{t-1})$ variables, as well as some latent variables $h_t$:

$$y_t = \boldsymbol{f}(\mathbf{x}_t) + \varepsilon_t^{\boldsymbol{u}} \quad \text{with;} \quad \varepsilon_t^{\boldsymbol{u}} \sim \mathcal{N}(0, \sigma_t^u(\mathbf{x}_t, h_t)) \quad ; \quad \mathbf{x}_t = \{\mathbf{c}_t, Y_t^{past}\},$$

with $\boldsymbol{f}(\mathbf{x}_t)$ the average explainable signal, and $\varepsilon_t^u$ a time-dependent Gaussian noise (local homogeneity assumption). The latter is associated with upstream irreducible variability, encompassing both intrinsic, measurement noises and premodeling noise arising from limits of the modeling scope (e.g. due to the influence of hidden variables $h_t$ that cannot be captured through lagged temporal variables $Y_t^{past}$).

The ML model $\hat{f}_\theta$ aims to approximate the explainable part $f$ of the target $y$ using observed variables $x$ from a training set $D_\theta = (\mathbf{x}_1, y_1), .., (\mathbf{x}_n, y_n)$, a subset of the dataset $\mathscr{D}$. $\theta$ is the set of parameters obtained using the training set $D_\theta$, over $\Theta$ indicating the whole set of parameters linked to all subsets of the dataset $\mathscr{D}$. According to the bias-variance trade-off (Eq.1), we decompose all error sources between $y_t$ and $\hat{f}_\theta(\mathbf{x}_t)$:

$$
\begin{aligned}
\boldsymbol{E}_{\boldsymbol{\Theta}}\left[(y_t - \hat{f}_\theta(\mathbf{x}_t))^2\right] &= \boldsymbol{E}_{\boldsymbol{\Theta}}\left[\hat{f}_\theta(\mathbf{x}_t) - f(\mathbf{x}_t)\right]^2 + \boldsymbol{E}_{\boldsymbol{\Theta}}\left[(\boldsymbol{E}_{\boldsymbol{\Theta}}\left[\hat{f}_\theta(\mathbf{x}_t)\right] - \hat{f}_\theta(\mathbf{x}_t)))^2\right] + E_y\left[(y_t - f(\mathbf{x}_t))^2\right] \\
&= \underbrace{\left(f_\Theta^*(\mathbf{x}_t) - f(\mathbf{x}_t)\right)^2}_{\text{Bias}} + \underbrace{\boldsymbol{E}_{\boldsymbol{\Theta}}\left[\left(f_\Theta^*(\mathbf{x}_t) - \hat{f}_\theta(\mathbf{x}_t)\right)^2\right]}_{\text{Variance}} + \underbrace{\sigma_t^u}_{\text{Intrinsic variability}},
\end{aligned} \tag{1}
$$

with $f_\Theta^* = \boldsymbol{E}_{\boldsymbol{\Theta}}[\hat{f}_\theta(\mathbf{x}_t)]$, the average function given the distribution $\Theta$. Among the three above-mentioned error sources, the *variance* can be explained by a noise $\varepsilon_t^\theta$ which corresponds to the gap between the average function over $\Theta$ and the ML model: $\varepsilon_t^\theta = f_\Theta^*(\mathbf{x}_t) - \hat{f}_\theta(\mathbf{x}_t)$. This epistemic noise is related to insufficient observations and could be reduced by gathering more data. The *bias* requires another random variable $\varepsilon_t^\Theta$ linked to the gap between the average explainable signal and the average function over $\Theta$:

$\varepsilon_t^\Theta = f(\mathbf{x}_t) - f_\Theta^*(\mathbf{x}_t)$. This noise, due to the modeling constraint over $\Theta$, is irreducible in the modeling scope. Finally, the *intrinsic variability* is related to the irreducible noise $\varepsilon_t^u$ that appears upstream of the modeling scope. It quantifies a lower bound for the expected error in the test data with both infinite data and unconstrained modeling. To show the relation between the introduced random variables and the epistemic/aleatoric concepts, we inject them into the total uncertainty law [8] in Eq.2. After simplification allowed by strong independence and zeros-mean assumptions:

$$\text{With } y_t = f_\theta(\mathbf{x}_t) + \varepsilon_t^\theta + \varepsilon_t^\Theta + \varepsilon_t^u \quad \text{and} \quad \sigma(y_t|x_t;\theta) = \sigma_\Theta\left[E_y(y_t|x_t;\theta)\right] + E_\Theta\left[\sigma_y(y_t|x_t;\theta)\right] = \boldsymbol{\sigma}_t^E + \boldsymbol{\sigma}_t^A$$

$$\text{We obtain} \quad \boldsymbol{\sigma}_t^E = \sigma_\Theta\left[\widehat{f}_\theta(\mathbf{x}_t)\right] = \boldsymbol{E}_\Theta\left[\left(\varepsilon_t^\theta\right)^2\right] \quad , \quad \boldsymbol{\sigma}_t^A = \sigma_y(\varepsilon_t^u) + \sigma_y(\varepsilon_t^\Theta)$$

(2)

From these equations, we can see that the decomposition into epistemic and aleatoric components (denoted by $E$ and $A$ superscripts) requires the manipulation of the whole set of parameters $\Theta$. As expected, the epistemic part is essentially made up of the variance error caused by the sampling of the training set. However, the aleatoric part contains several quantities that are all irreducible in the modeling scope but may be associated with different sources: upstream modeling scope (intrinsic, measurement, and pre-modeling noise), and model constraints which also cause bias. When we move slightly outside the domain of validity of the assumptions (due to limited training data and approximate manipulation of $\Theta$), the previous negligible terms can then induce blurs into the uncertainty decomposition.

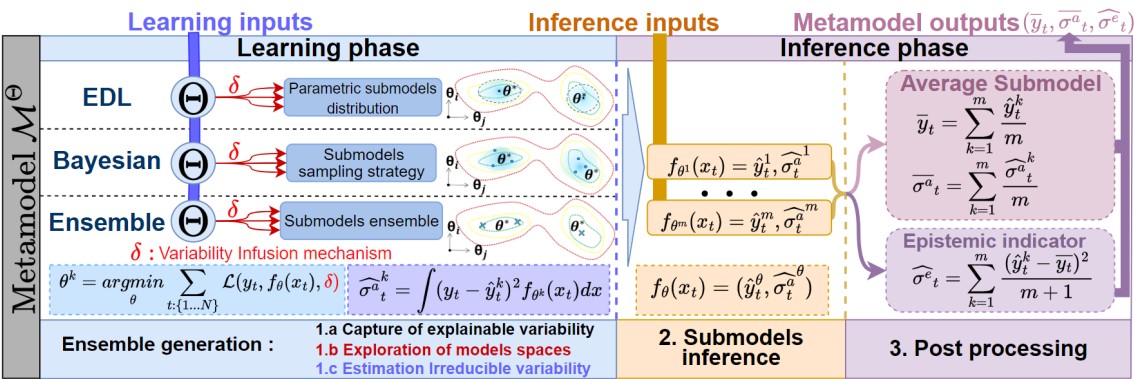

Figure 1: Illustration of a metamodel using Gaussian Aleatoric and Epistemic assumptions.

**Proposed unified dUQ framework:** The functional scheme of the proposed dUQ framework incorporating various UQ paradigms is shown in Fig. 1. It is based on a metamodel $\mathcal{M}^\Theta$ that learns and manipulates diverse submodels ($\hat{f}_\theta$) to combine their inferences. The learning phase aims to capture the explainable variability and estimate irreducible variability while exploring a diversity of submodel candidates $\Theta$. To ensure diversity and avoid submodel redundancy, a variability infusion mechanism (depending on the UQ paradigm) is needed during the learning phase. The estimated submodels produce, at the inference step, a local regressor $\hat{f}_\theta(\mathbf{x}_t)$ and an estimation of aleatoric variability $\hat{\sigma}^a(\hat{f}_\theta(\mathbf{x}_t))$. Furthermore, an epistemic variability $\hat{\sigma}_t^e$ is produced by computing the variability of the submodel regression $\hat{y}_t$ (using for example a Gaussian

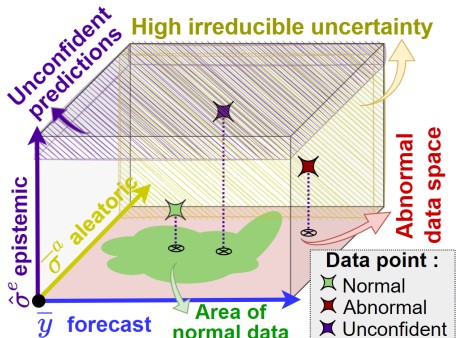

Figure 2: Theoretical dUQ indicators space

assumption). Finally, the metamodel provides a *risk-aware forecast* comprising three indicators: $\overline{\mu}_t, \overline{\sigma^a}_t, \widehat{\sigma^e}_t$ expressing forecast, aleatoric and epistemic indicators. As can be seen in Fig. 2, these indicators correspond to three independent axes on how the model perceives the data regarding forecast and sources of uncertainty. We can use them to design confidence intervals, error margins, or warnings highlighting a lack of model confidence.

## 3   PROPOSED DISENTANGLED UQ EVALUATION METHODOLOGY

Our dUQ evaluation methodology aims to perform a robust evaluation of model epistemic confidence, despite the absence of ground truth in real datasets. As an evaluation criterion, to consider dUQ limit due to modeling approximation, we propose the *disentangled Epistemic indicator* (dE-Ind) corresponding to a Negative Epistemic Ratio under Total Log-Likelihoods. It is computed through Aleatoric and Epistemic Gaussian assumptions from dUQ model output $(\overline{\mu}_t, \overline{\sigma}^a_t, \widehat{\sigma}^e_t)$ as $\qquad I^e_t = -ln(1 + \frac{\widehat{\sigma^a}_t}{\overline{\sigma^e}_t})$   (dE-Ind)

The experimental goal is to highlight an epistemic confidence gap in model predictions, between nominal and altered queries (i.e. affected by injections of *epistemic uncertainty*). Epistemic variability injections are designed to force the metamodel to extrapolate predictions on altered queries, corresponding to potentially unseen or even inconsistent feature space locations. In this latter case, the temporal correlation between features holding complex dependence on each other is potentially broken [14] and corresponds to out-of-distribution data. A distinction can then be made between altered queries close to the training domain boundary (almost-normal instance) and altered queries far outside the training domain boundary (abnormal instance). In what follows, we describe our methodology, which consists of two types of epistemic variability injections (Fig. 3), one at the *inference step* and another at the *training step* leading to different levels of experimental complexity and realism.

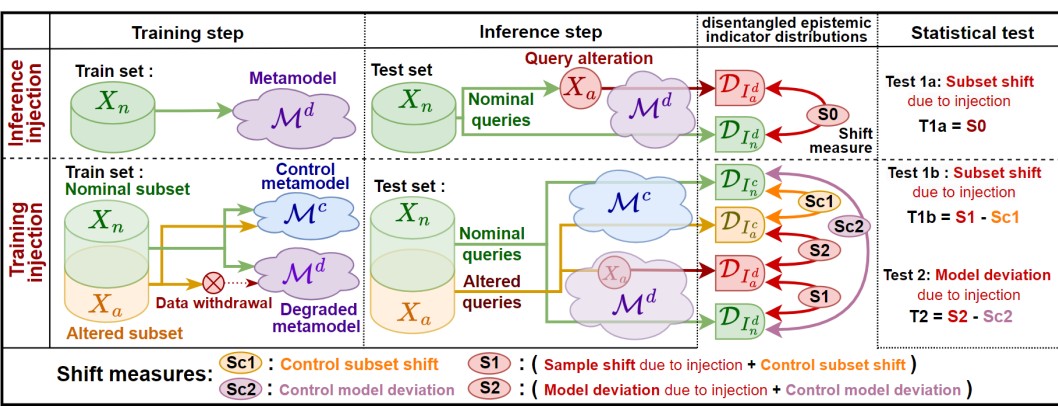

Figure 3: Scheme of the proposed dUQ evaluation methodology.

**Inference step injection** uses a data replacement to form a kind of robustness attack. To produce altered queries, the most important features are identified (according to the SHAP and SAGE libraries [7]) and their values are replaced by outliers belonging to distribution tails. Data replacements are done using quantile feature distributions on the whole dataset (global) or on a subset of data (local). The number and type of replacements determine the characteristics of the variability injection. Hence, computing a pre-trained metamodel $\mathcal{M}^d$ predictions on nominal ($X_n$) and altered ($X_a$) queries allow us to *statistically* quantify the epistemic confidence gap due to performing inference on naive synthetic outliers.

**Training step injection** uses a data withdrawal approach. In addition to the pre-trained metamodel (called *control*), a second instance of the same metamodel (called *degraded*) is trained on a slightly modified dataset: a selected subset (of neighboring data in the feature space) is ablated from training data (called *altered subset*) by a large portion (98% or 100%). Therefore, a distinction is made between nominal queries $X_n$ (which belong to *unaltered* subsets) and altered queries $X_a$. Here, the epistemic confidence gap is quantified by comparing the control ($\mathcal{M}^c$) and degraded ($\mathcal{M}^d$) model predictions on the test parts of the nominal and altered queries. As it is a more complex setup generating more realistic outliers, we designed a robust methodology based on statistical tests. These tests are corrected by a control mechanism accounting for both the shift between control and degraded models (due to divergence during learning) and the original shift between subsets (due to training set heterogeneity).

Both experimental setups aim to investigate whether the injection of epistemic variability induces a significant shift between the distributions of the dE-Indicator $\mathcal{D}_I$ predicted by a metamodel for nominal queries $F(\mathbf{x}_n) \sim \mathcal{D}_{I_n}$ and for altered queries $F(\mathbf{x}_a) \sim \mathcal{D}_{I_a}$. For the training injection setup, two statistical tests allow quantifying the significance of the dE-Indicator distribution shift with two distinct control measures. The first test, *Model deviation due to injection corrected by the control model deviation (test 1b in Fig. 3)*, highlights an epistemic confidence gap between the control and the degraded model for the altered subset, with a more substantial magnitude than for the nominal subset. The second test, *Sample shift due to injection corrected by the control subset shift (test 2)* highlights an epistemic confidence gap between the altered and nominal subsets for the degraded model, with a stronger magnitude than the original gap for the control model. Our statistical framework (see Section D of Appendix) is based on Wilcoxon-Mann-Whitney and Wilcoxon signed rank tests, with the hypothesis $H_0$ that the two distributions are identical and the alternative hypothesis that $\mathcal{D}_{I_a}$ stochastically dominates $\mathcal{D}_{I_n}$.

## 4 EXPERIMENTAL SETTINGS AND RESULTS

The benchmark aims to compare the performances of four metamodel-based approaches from different UQ paradigms applied to univariate time series: Random Forest disentangled Uncertainty Quantification (RF-dUQ)[26], Probabilistic Neural Network Monte Carlo Dropout (PNN-MCDP) [11], Probabilistic Neural Network Deep Ensemble (PNN-DE)[17] and Evidential Deep Learning regression (EDL)[3]. Implementations and datasets are available in Github repository. The developments are performed using standard ML libraries (Scikit-Learn [22] and Tensorflow[1]) on CPUs (see Section F of Appendix).

Table 2: Test set performance of *nominal* setup using our two forecasting datasets.

| Approach | MLP | RF-dUQ | PNN-MCD | PNN-DE | EDL | RF-dUQ | PNN-MCD | PNN-DE | EDL |
|---|---|---|---|---|---|---|---|---|---|
| Dataset | | RMSE metrics (lower is better) | | | | NLL metrics* (lower is better) | | | |
| real | $0.22_{\pm 0.02}$ | $0.23_{\pm 0.02}$ | $0.22_{\pm 0.2}$ | $\mathbf{0.22}_{\pm 0.02}$ | $0.22_{\pm 0.01}$ | $-0.51_{\pm 0.06}$ | $-0.55_{\pm 0.08}$ | $\mathbf{-0.57}_{\pm 0.07}$ | $-0.55_{\pm 0.08}$ |
| synthetic | $\mathbf{0.43}_{\pm 0.01}$ | $\mathbf{0.43}_{\pm 0.01}$ | $0.44_{\pm 0.01}$ | $0.43_{\pm 0.01}$ | $0.44_{\pm 0.01}$ | $0.43_{\pm 0.01}$ | $0.46_{\pm 0.01}$ | $\mathbf{0.40}_{\pm 0.02}$ | $0.44_{\pm 0.01}$ |
| Dataset | | Sharpness* | | | | Coverage (Target: 95.65%) | | | |
| real | $\varnothing^*$ | $0.82_{\pm 0.01}$ | $0.81_{\pm 0.02}$ | $\mathbf{0.73}_{\pm 0.01}$ | $0.75_{\pm 0.02}$ | $94.9_{\pm 0.8}$ | $94.9_{\pm 1.3}$ | $95.1_{\pm 1.4}$ | $94.4_{\pm 1.7}$ |
| synthetic | $\varnothing^*$ | $1.78_{\pm 0.01}$ | $1.86_{\pm 0.05}$ | $\mathbf{1.56}_{\pm 0.03}$ | $1.80_{\pm 0.03}$ | $96.7_{\pm 0.1}$ | $96.3_{\pm 0.1}$ | $95.0_{\pm 0.01}$ | $96.5_{\pm 0.2}$ |

*NLL, Coverage and sharpness is meaningless for the MLP model.

Firstly, we compare performances of the mentioned methods in terms of regression accuracy (Root-Mean-Square Errors, RMSE) and UQ relevance (Negative Log Likelihood, NLL) with six literature approaches over four public datasets [1] (see Table 2). We observe good overall performances of all methods, with slightly better results obtained by *Autodeuq* [10] (improved PNN-DE using AutoML).

Hereafter, we perform our dUQ evaluation on two new forecasting datasets. A real dataset[1] related to gas demand prediction and a synthetic one[1] based on local time-dependent Gaussian distribution sup-

---

[1] More information about methodology, and additional results are provided in Section E of Appendix.

port. The evaluation process[1] takes place in a standard ML framework with a sequential cross-validation scheme (2-folds and 2 repetitions) to ensure the robustness of the results. It is divided into 3 steps:

**1. UQ regression evaluation on two datasets**   We evaluated our four approaches on a nominal setup (without any data alteration) to ensure comparable accuracy and calibrated variance. It is done globally and locally (on several homogeneous subsets) to better acknowledge the data heterogeneity effects. We consider two additional metrics to evaluate the relevance of UQ: Sharpness[1] and Coverage[1] (i.e. size and % of data in the confidence interval respectively). Table 2 shows the competitive performance in terms of regression accuracy obtained by UQ-based approaches compared to a simple Multi-Layer Perceptron (MLP) model without UQ. Each approach obtains a coverage close to the theoretical one, although PNN-DE seems to provide narrower confidence intervals for similar coverage.

Table 3: Comparison of UQ regression performances using RMSE and NLL metrics on public datasets.

| Approach | PBP | MC Dropout | Deep Ens | hyper Ens | DF Ens | A-deuq | RF-dUQ | PNN-MCDP | PNN-DE | EDL |
|---|---|---|---|---|---|---|---|---|---|---|
| | | | Litterature performance | | | | Our metamodels | | | |
| Dataset | | | | | RMSE metrics (lower is better) | | | | | |
| **Kin8nm*** | 0.1 | 0.1 | 0.09 | $0.26_{\pm 0.0}$ | 0.09 | $\mathbf{0.06}_{\pm 0.0}$ | $0.142_{\pm 0.0}$ | $0.069_{\pm 0.0}$ | $0.067_{\pm 0.0}$ | $0.068_{\pm 0.0}$ |
| **powerplant*** | 4.12 | 4.02 | 4.11 | $4.38_{\pm 0.02}$ | 4.10 | $\mathbf{3.43}_{\pm 0.08}$ | $3.69_{\pm 0.13}$ | $3.75_{\pm 0.12}$ | $\mathbf{3.44}_{\pm 0.12}$ | $3.56_{\pm 0.15}$ |
| **protein*** | 4.73 | 4.36 | 4.71 | $5.09_{\pm 0.01}$ | 4.98 | $\mathbf{3.52}_{\pm 0.02}$ | $3.60_{\pm 0.03}$ | $3.77_{\pm 0.08}$ | $\mathbf{3.48}_{\pm 0.08}$ | $3.57_{\pm 0.05}$ |
| **yearprediction**** | 8.88 | 8.85 | 8.89 | $16.84_{\pm 0.08}$ | 9.30 | $\mathbf{7.91}_{\pm 0.04}$ | 9.25 | 8.75 | $8.71_{\pm 0.0}$ | 8.9 |
| Dataset | | | | | NLL metrics (lower is better) | | | | | |
| **Kin8nm*** | -0.9 | -0.95 | -1.2 | $6.89_{\pm 2.85}$ | -1.14 | $\mathbf{-1.40}_{\pm 0.01}$ | $-0.538_{\pm 0.02}$ | $-1.33_{\pm 0.01}$ | $-1.33_{\pm 0.01}$ | $-1.303_{\pm 0.02}$ |
| **powerplant*** | 2.84 | 2.8 | 2.79 | $5.24_{\pm 0.72}$ | 2.83 | $2.66_{\pm 0.05}$ | $2.69_{\pm 0.01}$ | $2.64_{\pm 0.01}$ | $\mathbf{2.55}_{\pm 0.02}$ | $\mathbf{2.55}_{\pm 0.02)}$ |
| **protein*** | 2.97 | 2.89 | 2.83 | $21.12_{\pm 2.52}$ | 3.12 | $2.48_{\pm 0.03}$ | $2.50_{\pm 0.01}$ | $2.351_{\pm 0.05}$ | $\mathbf{2.06}_{\pm 0.06}$ | $3.23_{\pm 0.10}$ |
| **yearprediction**** | 3.6 | 3.59 | 3.35 | $7.44_{\pm 0.08}$ | 3.58 | $\mathbf{3.22}_{\pm 0.00}$ | 3.64 | 3.31 | **3.22** | 3.30 |

*Cross-validation with 5-fold          **No cross validation due to the size of dataset

**2. Detailed dUQ evaluation on a training injection experiment on real data**   We propose an experiment based on three subsets of the real dataset sharing homogeneous characteristics in terms of their variance (see Appendix E for details): *low-variability* subset, *mid-variability* subset and *high-variability* subset. We present the detailed results in Fig. 4 for a training variability injection with the withdrawal of 98% of the mid-var subset. For each approach, performances of the control and degraded models (denoted by $c$ and $d$ respectively) are shown for each subset. Control meta-models of each of the four approaches display similar behaviors through all the metrics and subsets. As expected, models make more errors on the high-var subset and their predictions are less confident (higher NLL), but still offer a satisfying coverage thanks to the local uncertainty estimation.

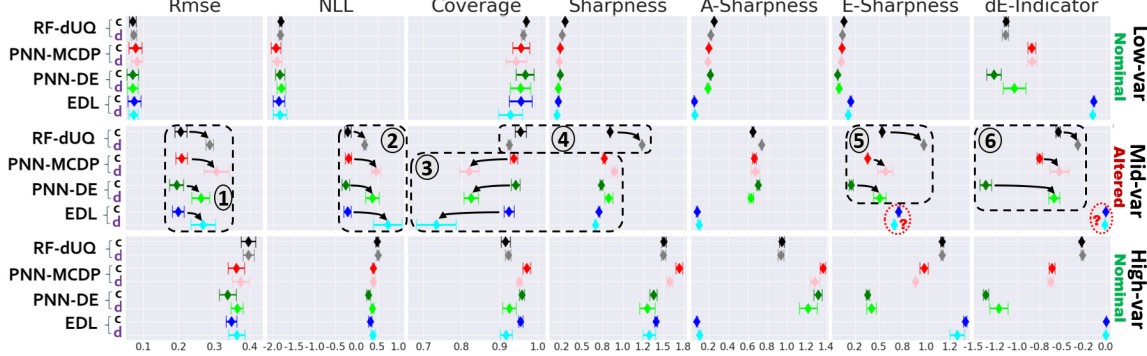

Figure 4: Local performances for one learning injection setup on real data. Control and degraded models are denoted by $c$ and $d$. Data are partitioned in three subsets of Low, Mid and High-variability.

By comparing control and degraded models, we observe their equivalent performances on nominal subsets (proof of injection locality). On the contrary, for the altered subset, the injection of variability leads to a loss of accuracy (arrows 1) and an increase in NLL (arrows 2), reflecting a loss of confidence between the degraded and control models. We observe a decrease in coverage (arrows 3) with a slight increase of sharpness for PNN-MCDP, PNN-DE, and EDL, meaning that the altered subset deviates from the nominal distribution. However, for RF-dUQ (arrows 4), we observe instead a sharpness increase.

The dUQ evaluation is performed through aleatoric/epistemic sharpness and the dE-Indicator (Fig. 4). Again, the control and degraded models show equivalent performances on nominal subsets. However, for the altered subset, all the approaches (except EDL) display a significant increase in epistemic sharpness (arrows 5) while there are only slight variations in aleatoric sharpness. The dE-Indicator logically increases (arrows 6), expressing a loss of epistemic confidence due to epistemic uncertainty injection. However, EDL shows no dE-Indicator increase, suggesting dUQ ineffectiveness in this case.

Finally, we represent the degraded PNN-DE model outputs in the UQ indicator space (introduced in Fig. 2), where each sample is a point whose coordinates are its three predictive indicators. On the left plot (colored by subset), green points (corresponding to the mid-var altered subset) are positioned higher on the epistemic axis, expressing the degraded model's lack of confidence at the inference step. The right plot (colored by epistemic confidence difference $\Delta dE$ between control and degraded models) proves that contrary to the degraded model, the control model doesn't express any lack of confidence in the mid-var subset. Indeed, the mid-var region involves higher $\Delta dE$ values, showing the fact that the observed lack of epistemic confidence is caused by mid-var data withdrawal.

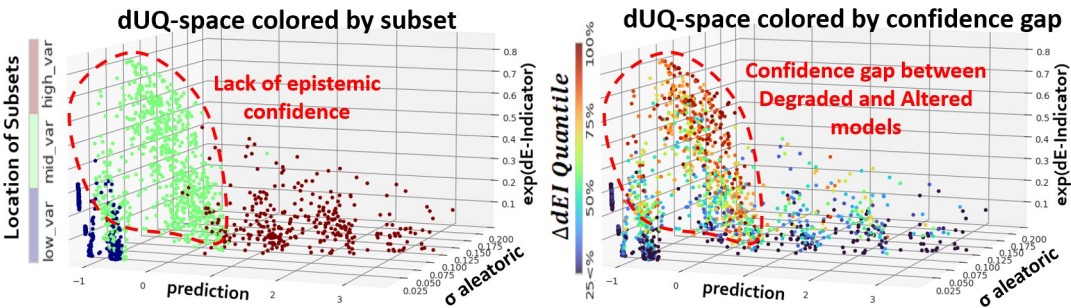

Figure 5: UQ space for PNN-DE degraded model on real data with mid-var data as altered subset, colored by variability-subset (left) and confidence gap $\Delta dE$ (right).

**3. Synthesis of evaluation of dUQ effectiveness on all experiments** In total, including cross-validations, 64 variants of experiments were performed using both injection setups on the real and synthetic data. Using the statistical framework based on dE-Ind distribution shifts, the objective is to determine whether epistemic injections may affect the epistemic component and whether their impact is significant on the aleatoric component.

For inference step injection experiments (test T1a of Fig. 6), all the approaches successfully expressed the lack of epistemic confidence (large margin above the red dotted line) presence of naive outliers. We observe the small impact of the injection strength for all approaches, whereas the type of injection (local vs. global) does not seem to have a significant impact. For training injection experiments, where a positive result must be observed for both tests (T1b & T2 in Fig. 6) to prove dUQ effectiveness, the results are more contrasted between the approaches. PNN-DE and PNN-MCDP show successful results in almost all configurations. RF-dUQ fails on the high-var setup. EDL fails in almost all configurations, illustrating that dUQ is not effective, either due to the intrinsic behavior of the approach or to parameterization issues in spite of hyperparameter optimization. The perturbation of the low-variability subset (low-var-98

and low-var-100) leads to small test scores for all approaches, suggesting difficulties in expressing low confidence in small variability data, even with few observations. However, some approaches (e.g. PNN-MCDP) still manage to express a loss of confidence even on low-magnitude injection. We also note that the training injection strength (98% vs. 100% removal) does not have a significant impact on dUQ effectiveness. A potential explanation relies on the fact the withdrawn samples have non-removed neighbors that retain part of the supporting information for prediction.

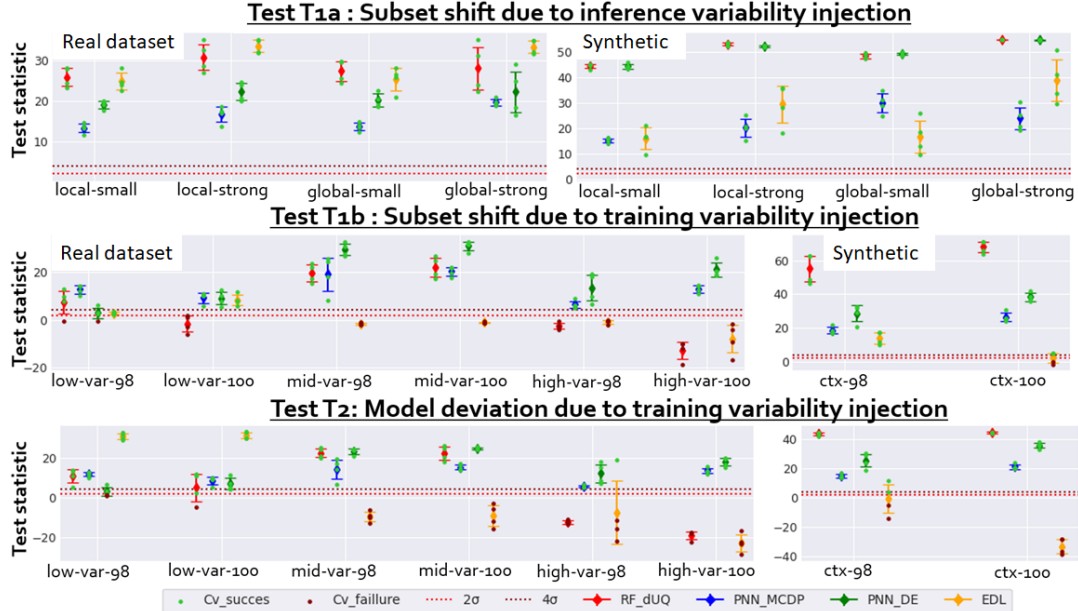

Figure 6: Results of the statistical tests for all experiments.

## 5 CONCLUSION AND PERSPECTIVES

We propose a dUQ evaluation methodology, based on epistemic injection at training or inference. These two mechanisms are designed to face methodological issues concerning the assessment of epistemic confidence without ground truth on real data. Experiments, performed using four state-of-the-art models and two datasets, demonstrate dUQ relevance and effectiveness on heterogeneous and heteroscedastic data. We show that some models (RF-dUQ, PNN-MCDP, PNN-DE) produce relevant local aleatoric and epistemic indicators on both datasets and succeed in handling naive altered queries. In contrast, others (EDL) show limitations when dealing with trickier outliers, resulting in ineffective dUQ.

**Limitation and perspectives** The current study only considers the regression task on time series, with Gaussian assumption of aleatoric and epistemic uncertainties. However, the extension to classification and other types of data is straightforward and within our perspectives. Future works will consider more complex and massive datasets issued from dynamic systems. Moreover, we aim to include more complex architectures (e.g. LSTM and Transformers) in our framework, go beyond the Gaussian assumption of UQ formalism, and compare more UQ paradigms using our framework (e.g., Bayesian and variational).

**Broader impact** To implement trustworthy AI in operational conditions, the risk-aware UQ framework will be a crucial part of a reliable chain combining control and certification mechanisms. It could be used along with data-qualification frameworks to ensure dataset viability and meet operational needs, such as complex systems monitoring, or anomaly and distribution drift detection.

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
