# EVALUATION METHODOLOGY FOR DISENTANGLED UNCERTAINTY QUANTIFICATION ON REGRESSION MODELS – APPENDIX

## CONTENTS

## A  INTRODUCTION

This document is the supplementary material of our paper entitled *Disentangled Predictive Uncertainty framework for Time Series Data*. We try to bring more details concerning various sections and experiments of the main paper when required. It follows the same structure as the main paper to facilitate its usage.

## B  RELATED WORKS

Predictive uncertainty estimation is a growing field in Machine Learning. In this area, a part of the research highlights the importance of distinguishing between two sources of uncertainty (popularized by Hüllermeier & Waegeman (2021)), and their consequences in reliability assessment and risk-based decision-making are discussed in Der Kiureghian & Ditlevsen (2009). Multiple survey papers on predictive *Uncertainty Quantification* (UQ) Abdar et al. (2021); Gawlikowski et al. (2021) propose classifications based on three main typologies of models: Bayesian inference, ensemble models, and evidential techniques. By analyzing the various techniques introduced in the state-of-the-art, we have identified a lack of cross-comparison and understanding. In what follows, we dedicate a section to provide a brief description of each typology alongside related works, and another section provides a comparative view of recent UQ approaches, motivating our contributions.

### B.1  UQ CATEGORIES

There has been a lot of interest around the Bayesian formalism to develop probabilistic methods for UQ. *Bayesian Neural Networks* (BNN) considers a prior probability distribution on the weights of the network, and the posterior distribution is computed during the training phase Blundell et al. (2015). As exact Bayesian inference is intractable for neural networks, a broad spectrum of approximations is proposed in the literature. Among these approximations, we can cite approximations based on sampling methods such as *Markov Chain Monte Carlo* (MCMC) Neal (2012), Laplace approximation Mackay (1992), and more recent research proposing *Monte Carlo DroPout* (MCDP) to estimate the uncertainty of forecast using Dropout in neural networks Gal & Ghahramani (2016); Zhu & Laptev (2017). The main advantage of the latest comes from the simplicity of its implementation in comparison to other approximations. Furthermore, the quality of the uncertainty estimation using classical approximations depends on the degree of the approximation and the correctness of the prior over the weights. The decomposition of uncertainty into epistemic and aleatoric indicators using a Bayesian framework is also suggested in the literature Depeweg et al. (2018); the authors suggest solutions based on both entropy and variance measures.

The *ensemble models* are widely used for uncertainty estimation due to their simple implementation. The uncertainty could be measured through the prediction confidence of the ensemble members. The well-known ensemble approach, namely, *Random Forests* (RF) can be used for the estimation of uncertainty indicators based on the total variance theorem Meinshausen & Ridgeway (2006); Shaker & Hüllermeier (2020). The ensemble formalism is also adapted to deep neural networks Lakshminarayanan et al. (2017). The proposed *Deep Ensemble* suggests learning a set of neural networks using the *Negative Log Likelihood* (NLL) criteria. It allows the estimation of both mean and variance over the data distribution through model disagreement.

Due to the similarity among the ensemble members in this approach, it may be limited when dealing with complex datasets generated from a combination of hypotheses. To alleviate this limitation and increase the diversity of models, various approaches are proposed in the literature Wen et al. (2020); Wenzel et al. (2020). These approaches concentrate on parameter sharing over ensemble members or

generating diverse members using a search over the space of hyperparameters. Another recent paper Egele et al. (2022) proposes to automate both the selection of members and their hyperparameters using *Neural Architectural Search* (NAS) and Bayesian optimization with the aim of increasing the diversity, predictive strength, and homogeneity of the ensemble models.

Based on the evidence theorem, *Evidential Deep Learning* (EDL) is proposed for the estimation of uncertainty indicators in a classification problem Sensoy et al. (2018). By learning a distribution over the parametric space of model outcomes, it collects evidence regarding the model predictions. Recently, the evidential formalism has been adapted for regression problems in Amini et al. (2020), while relaxing some prerequisites used for classification. A recent extension to generalize this technique for multivariate cases is proposed in Meinert & Lavin (2021). Despite the well-established probabilistic framework of EDL, they suffer from inaccurate predictions of target variables due to gradient shrinkage of the original loss function. An improvement over this problem using multitask learning is proposed in Oh & Shin (2022).

## B.2    CROSS COMPARISON

For the sake of completeness, we bring again our cross analysis in this section which makes use of the approaches described in the previous section (this section is the same as the section **??** in the main paper). We propose to analyze the recent literature through three main categories of characteristics: features, problem support, and environment configuration. Therefore, we provide a synthetic table (see Table 1) summarizing this information. Each main category includes multiple criteria, allowing the characterization of the studied methods. For the sake of completeness, we have added two additional approaches that focus on uncertainty estimation, which do not belong to identified UQ categories in section B of appendix. They are based respectively on Kalman filter Russell & Reale (2021) and direct output *Prediction Intervals* (PI) using neural networks. Pearce et al. (2018).

From Table 1 of the main paper, we can observe the existence of a diverse range of techniques (from Bayesian to evidential), each having its own characteristics. Some consider the distinction between two sources of uncertainty in their modeling, whereas others output a unique quantity presenting the total uncertainty. The regression problem is mostly addressed, whereas the adaptation to classification is straightforward. Furthermore, most approaches are evaluated on small public or synthetic datasets with some limitations in the evaluation methodology. Firstly, the datasets do not reflect the underlying complexity of real-world data due to inherent noises from heterogeneous sources (human, inherent, missing knowledge, etc.). On t, there is a lack of interpretation and behavioral analysis of uncertainty components (in particular, with respect to the epistemic part) in the evaluation, which often focuses on a global NLL metric.

The influence and suitability of the evaluation metrics for UQ models are also studied in the literature Guo et al. (2017); Ashukha et al. (2020); Nicora et al. (2022). They proposed the use of reliability diagrams (or reliability based on the local fit and density principles) and calibrated measures for comparison of various approaches, without addressing the interpretability and cross-comparison of uncertainty measures.

This article used a common formalism, described in the next section, to compare various typologies of UQ models, with the aim of better understanding their inherent behavior. We also propose an dUQ evaluation methodology to characterize the effectiveness of uncertainty estimation models in the absence of ground truth.

# C  FORMALISM

## C.1  NOTATION

**NOTATIONS**

| Basic term | |
|---|---|
| $t$ | index associated with a time step of a time series |
| $k$ | index associated to a member (submodel) of a metamodel |
| $N$ | number of samples (time series) |
| $m$ | number of submodels in a metamodel |
| $\hat{a}$ | Estimation of the quantity a |
| $\overline{a}$ | Average of the quantity a (often used on a set of estimations) |
| **ML Modeling variable** | |
| $y_t$ | the value of the series $y$ (ground truth) at time step $t$ |
| $f$ | function expressing the explainable part of the series $y$ |
| $\boldsymbol{c}_t, \boldsymbol{h}_t$ | exogenous variables (observed, hidden) impacting the time series $y_t$ |
| $Y_t^{past}$ | lagged endogenous variables from series $y$ used for forecasting of future values |
| $\mathbf{x}_t$ | the input data at time stamps $t$ composed in our framework by $(\boldsymbol{c}_t, Y_t^{past})$ |
| $D, D_\theta$ | dataset composed of pairs $(X_1, y_1), .., (X_N, y_N)$, and subset of dataset used to train a model |
| $\Theta$ | Ensemble of parameters set |
| $\theta, \theta^i, \theta^*$ | parameters set (resp one, the $i^{th}$, the "optimal") from $\Theta$ |
| $\hat{f}_\theta(\mathbf{x}_t) = \hat{y}_t$ | a ML model (function approximation) parametrized by $\theta$ with $x$ as inputs and providing $\hat{y}$ as output |
| **UQ modeling term** | |
| $\delta$ | abstract diversity infusion mechanism allowing the exploration of $\Theta$ space (ex bootstrap). |
| $\mathcal{M}, \mathcal{M}^c, \mathcal{M}^d$ | metamodel, control metamodel, degraded metamodel impacted by a variability injection |
| $\varepsilon^u, \varepsilon^\Theta, \varepsilon^\theta$ | Noise (uncertainty) link respectively to upstream sources, modeling constraint, and epistemic issues |
| $\sigma_t^A, \sigma_t^E$ | Variance (Aleatoric, Epistemic), at step $t$ |
| **model and meta-model output** | |
| $\widehat{f}_\theta(\mathbf{x}_t) = (\hat{y}_t^\theta, \hat{\sigma}_t^{A,\theta})$ | Outputs (Forecast and aleatoric estimation) corresponding to submodel $\theta$ at time step $t$ |
| $M(\mathbf{x}_t) = (\overline{y}, \overline{\sigma}_t^A, \hat{\sigma}_t^E)$ | Outputs (Forecast, aleatoric, epistemic estimation) corresponding to metamodel M $\theta$ at time step $t$ |
| **Experimental modeling term** | |
| $I_t^e$ | disentangled Epistemic indicator (dE-Indicator) |
| $X_n, X_a$ | nominal query without changes, altered queries that are degraded by variability injection |

## C.2  UNCERTAINTY COMPONENTS ENTANGLEMENT INTERPRETATION

The decomposition of uncertainty into aleatoric and epistemic components is a particular framework that seeks to disentangle irreducible forecast uncertainty from model uncertainty Der Kiureghian & Ditlevsen (2009). Given a modeling scope $S = \{X, y, F\}$ where X represents the data, y the targets, and F the model typology, we mentioned that this decomposition can be interpreted as a way to distinguish two kinds of *ill-posed* ML problems:

- For a selected experimental ML framework *S*, the epistemic uncertainty component can be reduced by increasing the quantity and / or representativeness of training data Der Kiureghian & Ditlevsen (2009)Hüllermeier & Waegeman (2021).

- The aleatoric component appears due to the irreducible forecast variability in the ML framework *S* Der Kiureghian & Ditlevsen (2009)Hüllermeier & Waegeman (2021). In this case, uncertainty is not a consequence of the *trained* model's lack of knowledge of the *trained* model but originates from the *upstream design* of the modeling scope *S*. This component informs us about the relevance of *S*, which intrinsically cannot be captured by the epistemic component alone.

In this paper, we address epistemic uncertainty through the use of metamodels and the variability of their individual submodel output. In this way, with more training data, optimization of the metamodel loss function is expected to converge towards parameters $\theta^*$ among the ensemble of a set of parameters $\Theta$ that is reachable according to the type of function F and the data (X,y).

On the other hand, the lack of an observed variable in the available data will contribute to the aleatoric uncertainty. The act of selecting relevant features prior to the model training is a methodological decision where the practitioner draws the line between what will be considered informative and what will be discarded as noise (i.e. irreducible uncertainty) Der Kiureghian & Ditlevsen (2009). The consequences of this choice impact the subsequent separation between aleatoric and epistemic uncertainty. It indeed establishes, prior to any loss function optimization and weight adjustment, the extent to which uncertainty will be treated either as a potential lack of information (epistemic component) or accepted as an irreducible limitation to the predictive power of the selected ML-framework $S$ (aleatoric component).

The balance between the epistemic and aleatoric components is a process in which information emerges as "a cut across the fabric of uncertainty"Malaspina & Brassier (2018). In that sense, epistemic and aleatoric uncertainty are not absolute notions, but rather defined with respect to each other and coupled together. Indeed, the limit case of an infinite number of features requires an infinite amount of (representative) data to fit. What was formerly discarded as noise is now deliberately incorporated in the ML framework $S$ as useful data originating from a latent process. In such an ideal situation, both the epistemic and aleatoric uncertainty components can asymptotically vanish to zero.

### C.3 BIAIS-VARIANCE TRADE-OFF AND TOTAL UNCERTAINTY LAW:

Taking up the formalization introduced in the paper, we introduced three aleatoric variable that can each be associated with a specific source of errors in the Biais-variance trade off.

- $\varepsilon_t^u = y_t - f(\mathbf{x}_t)$ the noise upstream the modeling scope associated with intrinsic irreducible variability errors, that corresponds to the difference between the series Y, and its explainable part $f$ according to the modeling scope $S$.

- $\varepsilon_t^\Theta = f(\mathbf{x}_t) - f_\Theta^*(\mathbf{x}_t)$ the modeling-constraint noise, related to bias errors due to the exploration space $\Theta$, that correspond to the difference between the explainable part $f-$ of data and the optimal reachable function $f_\Theta^*$ in $\Theta$.

- $\varepsilon_t^\theta = f_\Theta^*(\mathbf{x}_t) - \widehat{f}_\theta(\mathbf{x}_t)$ the epistemic noise, related to variance errors, that correspond to the difference between the optimal reachable function $f_\Theta^*$ in $\Theta$, and the obtained model $\widehat{f}_\theta$ which is biased by limited dataset $D_\theta$.

This modeling frame provides us with an artificial way to decompose the series $y$ according to a model, and its different sources of errors :

$$\text{With:} \quad y_t = f(\mathbf{x}_t) + \varepsilon_t^u = f_\Theta^*(\mathbf{x}_t) + \varepsilon_t^\Theta + \varepsilon_t^u = \widehat{f}_\theta(\mathbf{x}_t) + \varepsilon_t^\theta + \varepsilon_t^\Theta + \varepsilon_t^u \tag{1}$$

This modeling frame provides us with an artificial way to decompose the series $y$ according to a model, and its different sources of errors. It can then be rejected in the total law of uncertainty decomposition to analyze how this term impacts an aleatoric versus epistemic decomposition :

$$\text{With}\quad \sigma(y_t|x_t;\theta) = \sigma_\Theta\left[E_y(y_t|x_t;\theta)\right] + E_\Theta\left[\sigma_y(y_t|x_t;\theta)\right] = \boldsymbol{\sigma}_t^{\boldsymbol{E}} + \boldsymbol{\sigma}_t^{\boldsymbol{A}}$$

$$\sigma_t^E \overset{\text{if ind.}}{=} \underbrace{\sigma_\Theta[E_y[\widehat{f}_\Theta^{\theta_k}(\mathbf{x}_t)]]}_{\boldsymbol{E}_\Theta\left[(f_\Theta^*(\mathbf{x}_t) - \hat{f}_\theta(\mathbf{x}_t))^2\right]} + \sigma_\Theta[E_y[\varepsilon_t^\Theta]] + \underbrace{\sigma_\Theta[E_y[\varepsilon_t^u]]}_{0^*} + \underbrace{\sigma_\Theta[E_y[\varepsilon_t^\theta]]}_{0^*}$$

$$\sigma_t^A \overset{\text{if ind.}}{=} E_\Theta[\underbrace{\sigma_y(f^{\theta_k}(\mathbf{x}_t))}_{0}] + \underbrace{\sigma_y(\varepsilon_t^\Theta)}_{modeling} + \underbrace{\sigma_y(\varepsilon_t^u)}_{upstream} + \underbrace{E_\Theta[E_y(\varepsilon_t^\theta - E_y[\varepsilon_t^\theta])^2]}_{0^*}$$

(2)

Two types of assumption allow making simplification. We can consider that the random variables are independent (ind.), obtaining independent terms, then consider that the different random variables are centered (for different reasons), which makes the terms negligible.

## C.4 DETAILED EXPLANATION OF THE dUQ FORMALISM :

**Submodels** are functions $\{\widehat{f}_\theta, with \theta \in \Theta\}$ parameterized by $\theta$ that allow the *local* estimation of a decision function and its irreducible (Aleatoric) variability $\varepsilon_t^{\boldsymbol{A}}$ around the time stamp $t$. We have seen that this quantity explains the inherent noise in the data, which could not be explained during the training of the models but could be measured at each time stamp $t$ for submodel $k$:

$$\varepsilon_t^{\boldsymbol{A}} \sim \mathcal{N}(0, \sigma_t^a): \quad \sigma_t^a = \int (y_t - f(x_t))^2 f(\mathbf{x}_t) dx \approx -\int (y_t - \widehat{f}_{\theta^k}(x_t))^2 \widehat{f}_{\theta^k}(\mathbf{x}_t) dx = \widehat{\sigma^a}_t^k. \quad (3)$$

For the Random Forest model, this variability can be treated as a neighborhood scheme Lin et al. (2002), allowing to approximate $f_{\theta^k}(\mathbf{x}_t)$ by introducing the neighborhood $N(\mathbf{x}_t)$ around the point $\mathbf{x}_t$. Such mechanisms take various forms in the literature (Bayesian model, Frequentist or Set-based approach) under different hypotheses (distribution assumption, moment estimation, quantiles, etc.). With a local Gaussian Aleatoric assumption justified by the application of central limit theorem on the hidden variables, our submodels estimate the conditional probability distribution of the target variable from the observed variables.

$$\theta^k = \operatorname*{arg\,min}_{\theta} \sum_{t=1}^{T} Loss(y_t, \widehat{y}_t^\theta, \widehat{\sigma^a}_t^\theta) \quad , \quad P_{\theta^k}(y_t|\mathbf{x}_t) \sim \mathcal{N}(\widehat{y}_t^k, \widehat{\sigma^a}_t^k)$$

**The metamodel** $\mathcal{M}^\Theta$ is composed of a submodel ensemble (or submodel family). To specify a well-constructed metamodel, we have to introduce an abstract set $\Theta$: the family of reachable functions (that can be approximated by the type of selected model) and relevant submodels regarding a standard ML framework $S$. Then, the metamodel generates, manipulates, and enhances this ensemble of submodels through four following steps:

**1. The generation of the submodels ensemble** is realized by introducing variability during parameter optimization $\theta$ through a variability infusion mechanism $\delta$, which takes different forms depending on the UQ paradigm. The mechanisms of our four evaluated approaches can be interpreted as bagging/bootstrap (Ensemble), local minimum convergence (Deep Ensemble), binomial probabilistic weight (MCDP), or parameterization evidence-based (EDL). Mathematically, this is often achieved by disturbing the learning process (eq. 4) to generate submodel variants. To avoid having redundant submodels (lack of diversity) or coarse estimations (lack of accuracy), the metamodel has to be well-defined meaning:

$$\{\mathcal{M}^{\Theta} = \mathcal{M}_{\{\theta^1,\dots,\theta^m\}} \text{ with } \theta^k \in \Theta \mid \theta^k = \underset{\theta}{\arg\min}\, Loss(Y, f_\theta(X), \delta_k)\} \quad \text{With} \tag{4}$$

- Considerable diversity of the submodels: $\forall (k, k')$ with $k \neq k', \quad d(\theta^k, \theta^{k'}) > \epsilon$
- Acceptable accuracy and generalization capacity of submodels: $\forall k \in [1, m], \; f_{\theta^k} \in \Theta$

**2. Average model extrapolation** could be virtually computed through the average decision and the average aleatoric indicator of the submodels ensemble in a Gaussian aleatoric assumption.

$$\overline{y}_t = \frac{1}{m} \sum_{k=1}^{m} \widehat{y}_t^k, \qquad \overline{\sigma^a}_t = \frac{1}{m} \sum_{k=1}^{m} \widehat{\sigma^a}_t^k \tag{5}$$

**3. Epistemic estimation** $\varepsilon_t^E$ allows for local estimation of uncertainty through the metamodel submodels. Epistemic confidence can then be interpreted as the likelihood of the *average submodel decision* ($\overline{y}_t$) over the metamodel. Under the well-formed ensemble assumption for a regression task, local decision estimators $f_{\theta^k}$ are centered with respect to the local mean $\mu_t$ and have independent errors. Then, a Gaussian epistemic assumption, thanks to unbiased (target-centered) models with independent errors, allows the use of the following unbiased empirical variance estimators.

$$\varepsilon_t^E \sim \mathcal{N}(0, \sigma_t^e): \qquad \sigma_t^e \approx \widehat{\sigma^e}_t = \sqrt{\frac{1}{m+1} \sum_{k=1}^{m} (\widehat{y}_t^k - \overline{y}_t)^2} \tag{6}$$

**4. Metamodel output** can express a decision and estimate its forecast uncertainty, $\mathcal{M}_{\{\theta^k\}_{k=1}^m}(\mathbf{x}_t) \sim \mathcal{N}(\overline{y}_t, \widehat{\sigma}_t)$ corresponding to the combination of aleatoric and epistemic uncertainties $\widehat{\sigma}_t = \overline{\sigma^a}_t + \widehat{\sigma^e}_t$. It should be noticed that there is a high correlation between the epistemic and aleatoric uncertainty indicators: the model error increases with high irreducible uncertainty. To better exploit local epistemic variability, we propose the dE-Indicator corresponding to a Negative Ratio of Epistemic under Total Log-Likelihoods through Aleatoric and Epistemic Gaussian assumptions: $\quad I_t^e = -ln(1 + \frac{\widehat{\sigma^a}_t}{\widehat{\sigma^e}_t})$

**additional remarks** Two additional remarks should be made. First, the *disentangled Uncertainty Quantification* (dUQ) framework may be extended with other Aleatoric and Epistemic assumptions (distribution law, quantile, other kinds of moments). These extensions will lead to three main issues: submodel aleatoric estimation, submodel aggregation, and epistemic extraction from the ensemble. Secondly, this framework "contains" the standard *Machine Learning* (ML) approaches if they are formalized under the constant aleatoric assumption and total confidence in the unique submodel. It also "contains" other classical ML UQ focusing on Aleatoric or Epistemic if formalized under the assumption that neglecting the other part.

## C.5 DE-INDICATOR DEVELOPMENT

By manipulating dUQ framework under assumptions about aleatoric and epistemic uncertainty, we can design predictive indicators from metamodel likelihood using the assumption A1: the submodels ensemble of metamodel is well-formed, which means rich in variability $\forall (k, k') \; with \; k \neq k' \quad d(\theta^k, \theta^{k'}) > \epsilon$ and composed of submodels with good predictions and with few overfitting $\forall \; k, \; f_{\theta^k} \in \Theta$. Starting from the normal log-likelihood using total variance $\sigma^{tot}$:

$$ln(L(\overline{y}_t, \widehat{\sigma^{tot}}_t; \Theta, y_t)) = ln(P(y_t; \overline{y}_t, \widehat{\sigma^{tot}}_t | \Theta)) \tag{7}$$

$$\approx ln(P(y_t; \overline{y}_t, \widehat{\sigma^{tot}}_t | \{\theta^k\}_{k \in [1,m]}))$$

$$= cst - ln(\widehat{\sigma^{tot}}_t) - \frac{(y_t - \overline{y}_t)^2}{2 * \widehat{\sigma^{tot}}_t}$$

*A1 submodel-ensemble approx*

*NLL developpement*

A local likelihood of the metamodel on the training set can be approximated under:

- the normal-aleatoric assumption $y_t \sim \mathcal{N}(\overline{y}_t, \overline{\sigma^a})$
- the normal-epistemic assumptions $\mu_t^* \sim \mathcal{N}(\overline{y}_t, \overline{\sigma^e})$
- the A2 assumption of a local unbiased capture based on the ability of the submodels to perform local estimation in a standard ML-framework.

We can derive the A2 assumption of a "local unbiased capture performed by submodels" by combining a standard ML-framework and normal-aleatoric assumption.

A2: Local neigboor observation approx: $y_t \approx \int \sum_k f_{\theta^k}(\mathbf{x}_t) dx \approx \int y_t dy$

$$ln(L(\overline{y}_t, \widehat{\sigma^{tot}}_t; \Theta, y_t)) \approx cst - ln(\widehat{\sigma^{tot}}_t) - \frac{(y_t - \overline{y}_t)^2}{2 * \widehat{\sigma^{tot}}_t} \tag{8}$$

$$\approx cst - ln(\widehat{\sigma^{tot}}_t) - \frac{(\int y_t dy - \overline{y}_t)^2}{2 * \widehat{\sigma^{tot}}_t}$$

$$\approx cst - ln(\widehat{\sigma^{tot}}_t) + 0$$

$$\approx cst - ln(\overline{\sigma^a}_t + \widehat{\sigma^e}_t) + 0$$

$$L(\overline{y}_t, \widehat{\sigma^{tot}}_t | \Theta)) \propto \frac{1}{\overline{\sigma^a}_t + \widehat{\sigma^e}_t}$$

*Formula 5 + $y_t \sim$ A2*

*$\overline{y}_t$: Local MLE by NLL loss minization*

A local epistemic likelihood of the "average-submodel" mean among the reachable and relevant models $\Theta$ can also be obtained under the A3 assumption of "local unbiased mean modelling approximation" stemming from the combination of the A1 assumption and normal-epistemic assumptions.

A3 Local mean modelling approx: $\mu_t^* \approx \int_\Theta \frac{\widehat{y}_t^k}{m} \approx \sum_{k=1}^m \frac{\widehat{y}_t^k}{m}$

$$ln(L(\overline{y}_t, \widehat{\sigma^e}_t | \Theta, \mu_t^*) = cst - ln(\widehat{\sigma^e}_t) - \frac{(\mu_t^* - \overline{y}_t)^2}{2 * \widehat{\sigma^e}} \tag{9}$$

$$\approx cst - ln(\widehat{\sigma^e}_t) - \frac{\left(\sum_{k=1}^m \frac{\widehat{y}_t^k}{m} - \overline{y}_t\right)^2}{2 * \widehat{\sigma^e}}$$

$$= cst - ln(\widehat{\sigma^e}_t) - 0$$

$$L(\overline{y}_t, \widehat{\sigma^e}_t | \Theta) \propto \frac{1}{\widehat{\sigma^e}_t}$$

*Formula 5 + $\mu_t^* \sim$ A3*

*$\overline{y}_t$: local MLE*

Finally, we are more specifically interested in the fluctuation of the local epistemic likelihood of the decision regarding local forecasting uncertainty to be more "invariant" to the magnitude of local forecasting uncertainty. Therefore, we propose to analyze the local epistemic likelihood indicator (Formula 9) normalized by the local total likelihood indicator (Formula 8).

$$\frac{L^{epi}(\overline{y}_t, \widehat{\sigma^e}_t | \Theta)}{L^{tot}(\overline{y}_t, \widehat{\sigma^{tot}}_t | \Theta)} \propto \frac{\overline{\sigma^a}_t + \widehat{\sigma^e}_t}{\widehat{\sigma^e}_t}$$

$$ln(\frac{L^{epi}}{L^{tot}}) \propto ln(\overline{\sigma^a}_t + \widehat{\sigma^e}_t) - ln(\widehat{\sigma^e}_t) \tag{10}$$

$$\propto ln(1 + \frac{\overline{\sigma^a}_t}{\widehat{\sigma^e}_t})$$

## D  STATISTICAL SETUP

**The proposed design of statistical test** aims to prove that the injection of epistemic variability induces a significant shift between the distributions of the *disentangled Epistemic indicator* (dEIndicator) $\mathcal{D}_I$ predicted by a metamodel for nominal queries $F(X_n) \sim \mathcal{D}_{I_n}$ and for altered queries $F(X_a) \sim \mathcal{D}_{I_a}$. To do so, we use Wilcoxon-Mann-Whitney test (or Wilcoxon signed-rank test for matched distributions), which compares two independent empirical distributions $X$ and $Y$ (with cumulative distributions $F$ and $G$) under the null hypothesis $H0 : P(X > Y) = P(X < Y)$, meaning that neither law is stochastically superior to the other. More precisely, we are interested in the rejection of $H0$ in favour of the alternative $H1 : P(X > Y) > P(X < Y)$, expressing the fact that the law of X is stochastically superior to the law of Y, i.e. for all $u$, $F(u) > G(u)$.

For the inference-injection setup, which is highlighted in the dashed red box in Fig. 1, dEIndicator distributions for nominal and altered subsets are paired (each altered query is created from a nominal one). So we can use Wilcoxon signed-rank test to prove:  $\mathcal{D}_{I_a} \underset{stochastically}{>} \mathcal{D}_{I_n}$.

Figure 1: Synthesis of both inference-injection and training-injection statistical experimental setups.

For the training-injection setup, Fig. 1 summarizes the design of the statistical tests. It can be interpreted as two Wilcoxon-Mann-Whitney tests (red cells) that quantify dEIndicator distribution shift, due to in-

jections corrected by control measurements for a **model-centered** comparison (bottom purple cell) or a **subset-centered** comparison (right purple cell).

1. **Model deviation due to injection corrected by control models deviation test**: this first test aims at highlighting a gap of epistemic confidence between control and degraded models on the altered subset, with a stronger magnitude than on the nominal subset. This entails evaluating whether dEIndicator distribution deviation between degraded and control models on altered queries (subset-shift proof) "dominates" the deviation on nominal queries (control).

2. **Sample shift due to injection corrected by the original sample shift test**: this second test aims to highlight a epistemic confidence gap between the altered and nominal subsets for the degraded model, with a stronger magnitude than the original gap for the control model (for which "altered" subset is, in this case, not altered). This entails evaluating whether dEIndicator the distribution of altered queries dominates the distribution of the nominal queries for the degraded model (data-shift proof) and whether this domination is stronger than the original one, evaluated through the control model (control).

## E  EXPERIMENTAL AND EVALUATION SETTINGS

**Datasets** The four metamodel approaches are evaluated for univariate time series forecasting with Aleatoric & Epistemic risk-awareness on two datasets[0]. A real one linked to gas demand prediction and a synthetic one based on local time-dependent Gaussian distribution support. Both datasets contain a set of categorical, lagged values, and other numerical observed variables (29 for the gas demand dataset and 20 for the synthetic one.

**Models** In this paper, we compare four standard approaches compatible with our dUQ framework. Tables 2 summarize the four metamodels characteristics (See supplementary materials for hyperparameters[0]):

1. ***Random Forest disentangled Uncertainty Quantification* (RF-dUQ)** *Ensemble*;
2. ***Probabilistic Neural Network Monte Carlo DropOut* (PNN-MCDP)**; *Probabilistic BNN*
3. ***Probabilistic Neural Network Deep Ensemble* (PNN-DE)**; *Deep-Ensemble*
4. ***Evidential Deep Learning regression* (EDL)**. *Parametric BNN*

Table 1: Comparison of the four models used in the dUQ framework.

| | Loss | Submodel | | | Metamodel | Metamodel Output | | |
|---|---|---|---|---|---|---|---|---|
| **Methods** | Loss | Type | Forecast | Aleatoric | Espitemic | Forecast | Aleatoric | Epistemic |
| **RF-dUQ** | MSE | Tree | regression | Set-based (Leaf) | Tree ensemble (explicit) | Mean of regression | Mean of Leaf-Var | Var of regression |
| **PNN-MCDP** | NLL | MLP PNN | $\mu$ (Gaussian) | $\sigma^2$ (Gaussian) | MCDP (virtual) | Mean of $\mu$ draws | Mean of $\sigma^2$ draws | Var of $\mu$ draws |
| **PNN-DE** | NLL | MLP PNN | $\mu$ (Gaussian) | $\sigma^2$ (Gaussian) | Deep ensemble (explicit) | Mean of $\mu$ set | Mean of $\sigma^2$ set | Variance of $\mu$ set |
| **EDL** | $\Gamma^{-1}$LL + reg | subpart of MLP | Mean-$\mathcal{N}$ law | $\sigma$-$\Gamma^{-1}$ law | Evidential (parametrized) | Mean of Mean-$\mathcal{N}$ | Mean of $\sigma$-$\Gamma^{-1}$ | Var of $\sigma$-$\Gamma^{-1}$ |

Note that neural network approaches are tested with a basic architecture of dense multilayer perceptron.

### E.1  METRICS

- Forecasting accuracy with Root-Mean-Square Error (RMSE): $\frac{1}{n}\sqrt{\sum_{t=1}^{n}(y_t - \overline{y}_t)^2}$

- UQ relevance with Negative Log Likelihood (NLL): $-\frac{1}{n}\sum_{i=1}^{n} -0.5 ln(2 * \pi) - ln(\widehat{\sigma^{tot}}_t) - \frac{(y_t - \overline{y}_t)^2}{2 * \widehat{\sigma^{tot}}_t}$

- UQ relevance with Sharpness (Average of $\beta$-LVL Confidence Interval size): $\frac{1}{n}\sum_{t=1}^{n} 2 * \beta * \sigma_t$

- UQ relevance with Coverage (% of data in the $\beta$-LVL Confidence Interval): $\frac{1}{n}\sum_{t=1}^{n} \mathbb{1}\{|r_t| \le \beta * \sigma_t\}$

- dUQ effectiveness with dEIndicator:  $I_t^e = -ln(1 + \frac{\widehat{\sigma^a}_t}{\sigma^e_t})$

### E.2  DATASETS

**Public dataset :** Before carrying out our experimental setup, 4 public datasets are used to ensure the relevance of our models. We chose univariate regression datasets having more than 5000 instances, among those used for the benchmarking of UQ approaches:

- CASP-protein.csv (~40000): dataset

- Kin8nm (~8000): dataset

- Folds5x2-pp (~10000): dataset

- Data-year-prediction (~500000): dataset

**Real dataset**  This dataset is based on weekly industrial gas deliveries across seven years. Each delivery corresponds to a combination of three categorical variables: the **source** of production, the **product** to be delivered, and the final **customer**. Having 29 possible deliveries, i.e. 29 combinations of sources, products, and customers overall, the dataset concatenates all these time series over the seven years, as illustrated in Fig. 2, amounting to around 10000 weekly observations in total. As there are heterogeneity and heteroscedasticity between the 29 subseries, it can be portioned into 3 classes of low, mid, and high variability in order to analyze data subset performances. Other numerical and categorical variables complete the historical consumption deliveries, e.g. customers-specific and order data, geographical distribution of sources and customers, as well as seasonal parameters. The incentive is to provide, for all deliveries, accurate gas demand predictions for the following week, by leveraging the last four weeks of data. Necessary steps of data anonymization and numerical transformations were also inevitable to ensure customers data confidentiality.

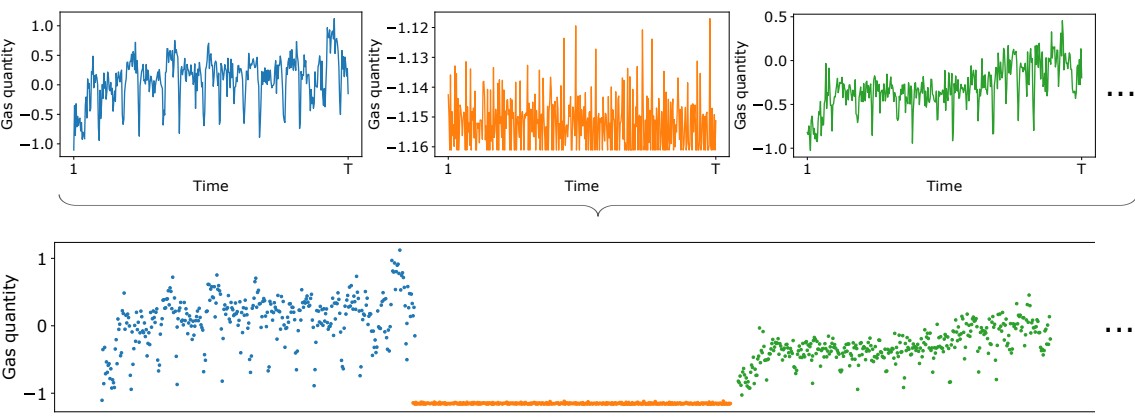

Figure 2: Visualization of the real gas-demand datasets.

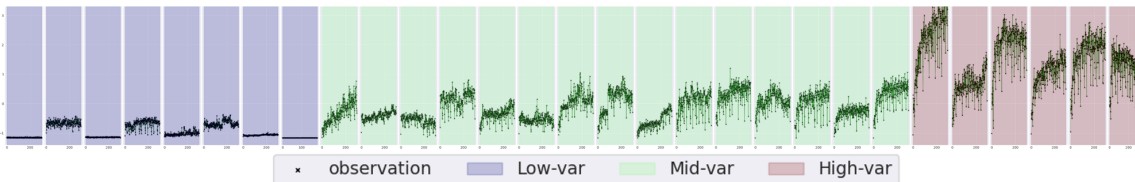

Figure 3: Visualization of the real gas-demand dataset, with colors corresponding to different variability subsets.

**Synthetic dataset** We also generate synthetic data which simulate time series with local time-dependent Gaussian distribution support, including white noise ($Y_t \sim \mathcal{N}(\mu(\mathbf{x}_t), \sigma(\mathbf{x}_t)) + \epsilon$). Mean and standard deviation parameters are independently structured by categorical, short-periodic, long-periodic, latent numerical, and autoregressive (time-dependent) features. The synthetic forecasting dataset contains 16000 observations with 16 observed variables. Fig. 4 illustrates a subpart (lower panel) as well as the whole (upper panel) synthetic dataset, with confidence intervals of the distribution support.

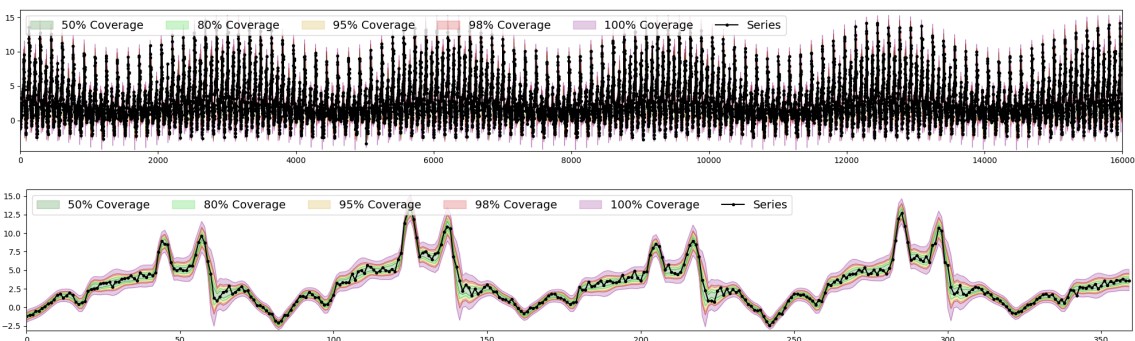

Figure 4: Visualization of the synthetic dataset.

**Cross-validation** All performance metrics are evaluated on a test set, unseen during training. To ensure robust evaluation while respecting the temporal integrity despite the small size of our data, we perform a duplicated 2-fold sequential cross-validation: the dataset is split into 3 parts in chronological order. Parts 1 and 2 correspond to the train and test sets of the first fold. Parts 2 and 3 correspond to the train and test of the second fold. Both folds are duplicates, leading to a total of 4 cross-validation sets. For experimental uncertainty injection setups (see the following section), we perform injections on each of the 4 cross-validation sets and store the resulting dataset to perform identical experiments for each approach.

**Models** In this paper, we compare four standard approaches compatible with our dUQ framework. Table 2 summarizes the four metamodels characteristics:

1. **RF-dUQ** *Ensemble*;

2. **PNN-MCDP**; *Probabilistic BNN*

3. **PNN-DE**; *Deep-Ensemble*

4. ***Evidential Deep Learning regression* (EDL)**. *Parametric BNN*

Table 2: Comparison of the four models used in the dUQ framework.

| Methods | Loss | Submodel | | | Metamodel | Metamodel Output | | |
|---|---|---|---|---|---|---|---|---|
| | Loss | Type | Forecast | Aleatoric | Espitemic | Forecast | Aleatoric | Epistemic |
| RF-dUQ | MSE | Tree | regression | Set-based (Leaf) | Tree ensemble (explicit) | Mean of regression | Mean of Leaf-Var | Var of regression |
| PNN-MCDP | NLL | MLP PNN | $\mu$ (Gaussian) | $\sigma^2$ (Gaussian) | MCDP (virtual) | Mean of $\mu$ draws | Mean of $\sigma^2$ draws | Var of $\mu$ draws |
| PNN-DE | NLL | MLP PNN | $\mu$ (Gaussian) | $\sigma^2$ (Gaussian) | Deep ensemble (explicit) | Mean of $\mu$ set | Mean of $\sigma^2$ set | Variance of $\mu$ set |
| EDL | $\Gamma^{-1}$LL + reg | subpart of MLP | Mean-$\mathcal{N}$ law | $\sigma$-$\Gamma^{-1}$ law | Evidential (parametrized) | Mean of Mean-$\mathcal{N}$ | Mean of $\sigma$-$\Gamma^{-1}$ | Var of $\sigma$-$\Gamma^{-1}$ |

Note that neural network approaches are tested with a basic architecture of dense multilayer perceptron.

Table 3: Metamodels parameters on real data.

| Parameters | Architecture | Variability infusion | Regularization* (submodel complexity) | Loss |
|---|---|---|---|---|
| RF-dUQ | Tree-depth: 18 | N-Tree: 50 max-features: 0.8 | ccp-alpha: 5e-5 impurity-dec:5e-5 | $\emptyset$ |
| PNN-MCDP | MLP: [200,300,200,2] f-acti: [ReLu,",",Linear] | Dropout: 0.25 n-draw: 6 | (L1-reg,L2-reg): (4e-4,4e-4) | log-var pen: 0.1.05 |
| PNN-DE | MLP: [200,300,200,2] f-acti: [ReLu,",",Linear] | n-model: 5 train-set: 85% | (L1reg,L2reg): (1e-3,1e-3) Dropout 0.02 | log-var pen: 0.74 |
| ED | MLP: [200,300,200,4] f-acti: [ReLu,",",edl] | $\emptyset$ | (L1reg,L2reg): (1e-3,1e-3) Dropout 0.02 | edl-reg: 15*e-2 |

The choice of hyperparameters was made manually in an iterative way after several experiments for the neural networks, and by grid search for the RF-dUQ. The models are implemented thanks to a code-overlay on scikit-learn (RF-dUQ) or in tensorflow for the deep learning models. The benchmark is implemented in python. The neural networks are optimized by a Nadam algorithm Dozat (2016). To guarantee a stable learning on all our experiments, we perform a learning procedure with three successive trainings by varying the batch sizes (128,32,64). Each training is limited to 1000 steps with two call back procedures aiming to reduce the learning rate in case of a learning plateau, and to interrupt the current training if the plateau is longer.

Table 4: Learning and inferences times for both real and synthetic datasets.

| Approach | Accuracy and execution times | | | | | UQ Performances** | | | | |
|---|---|---|---|---|---|---|---|---|---|---|
| | MLP | RF-dUQ | PNN-MCD | PNN-DE | EDL | MLP | RF-dUQ | PNN-MCD | PNN-DE | EDL |
| Dataset | Average training time (s) | | | | | Average inference time (s) for whole dataset | | | | |
| gas-demand | 217 | 12 | 614 | 1374 | 364 | 2.2 | 10.25* | 10.3 | 5.8 | 1.3 |
| synthetic | 224 | 9 | 641 | 2968 | 610 | 1.0 | 14* | 7 | 12 | 1.4 |

*Inference time is high due to unoptimized implementation ** UQ relevance is meaningless for the MLP model

### E.3 QUALITATIVE RESULTS OVERVIEW ON NOMINAL SETUP

Fig. 5 provides a qualitative view of PNN-MCDP output, allowing to observe good adaptability of confidence intervals sizes to subset variability. $\sigma$ and $2\sigma$ thresholds along with the epistemic confidence levels are extrapolated from the predictive uncertainty indicators.

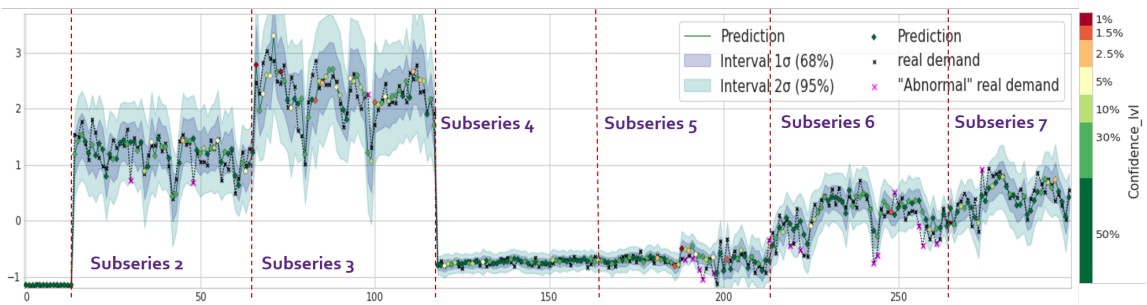

Figure 5: Visualization of t+1 prediction with UQ indicators for PNN-MCDP.

Figs. 6 and 7 provide qualitative views of the results of the 4 approaches for real and synthetic datasets.

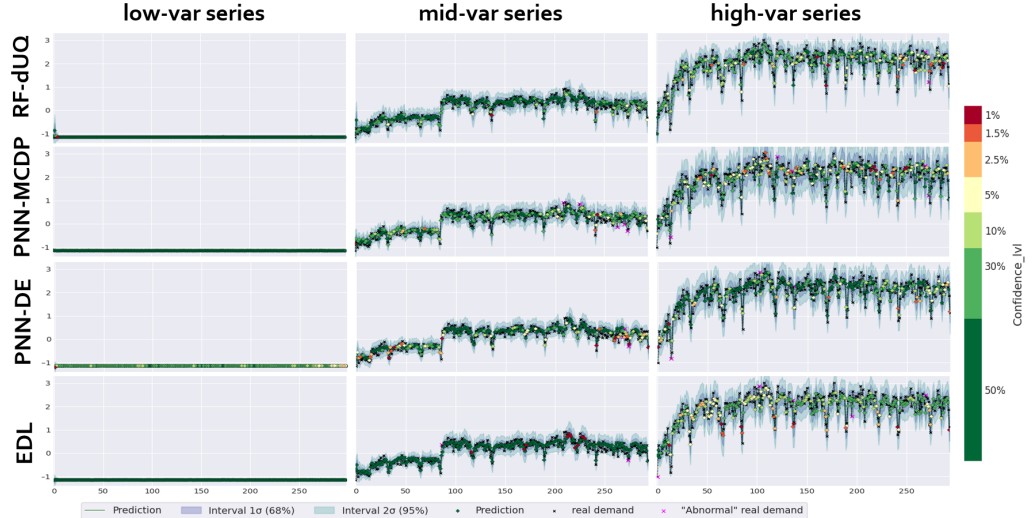

Figure 6: Visualization of t+1 predictions with UQ indicators of the 4 approaches on real data.

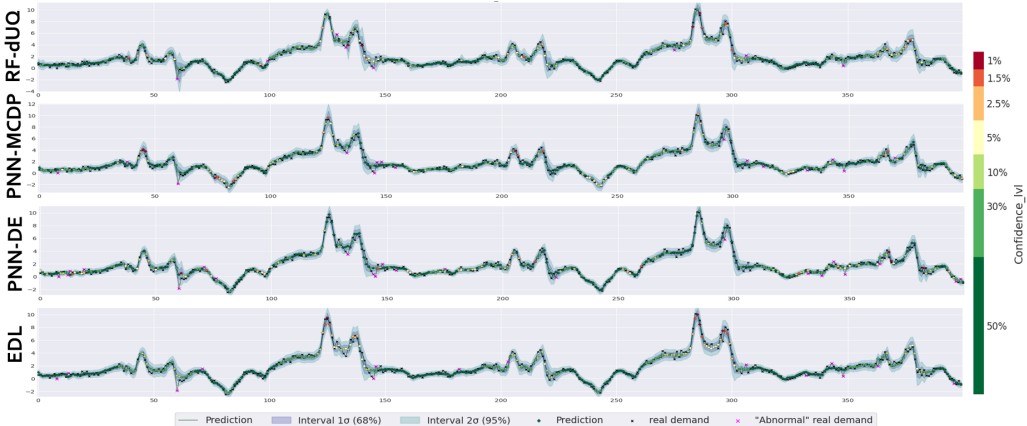

Figure 7: Visualization of t+1 predictions with UQ indicators of the 4 approaches on synthetic data.

### E.4 DUQ SPACES FOR THE 4 APPROACHES ON REAL DATA UNDERGOING THE PRESENTED LEARNING INFERENCE INJECTION

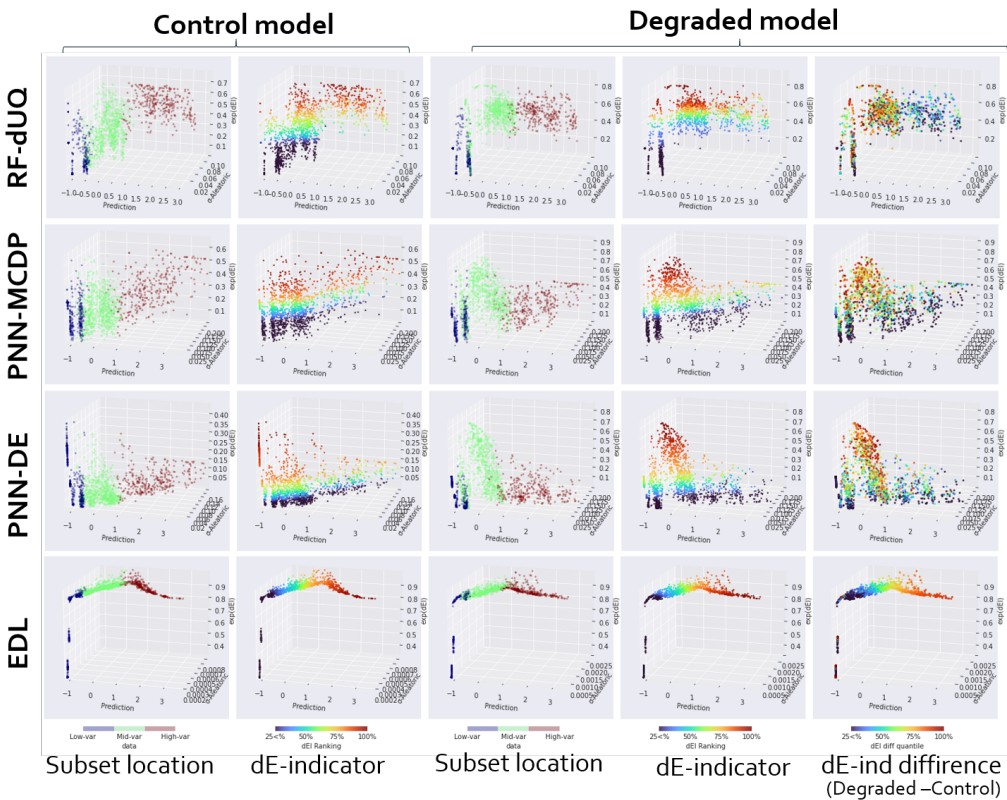

Figure 8: dUQ space for degraded and control versions of the 4 approaches on a real dataset, with learning variability injection on mid-var subset.

### E.5 DETAILED RESULTS FOR THE LEARNING VARIABILITY INJECTION ON SYNTHETIC DATA (WITH 100% DATA WITHDRAWAL)

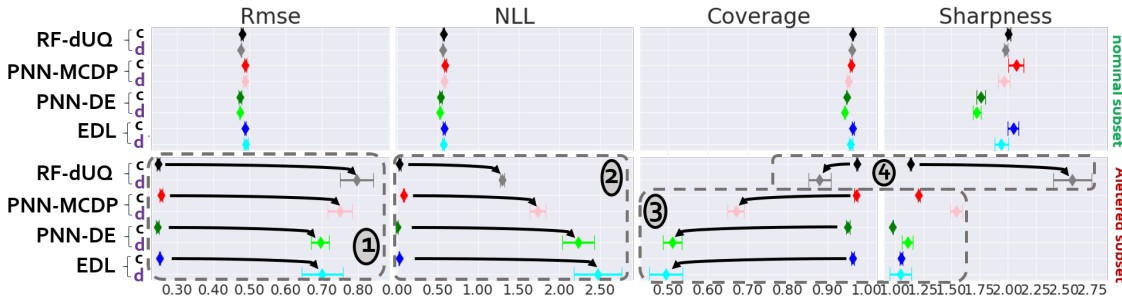

Figure 9: Performance metrics of the 4 approaches on nominal and altered subset, for a learning variability injection on synthetic data.

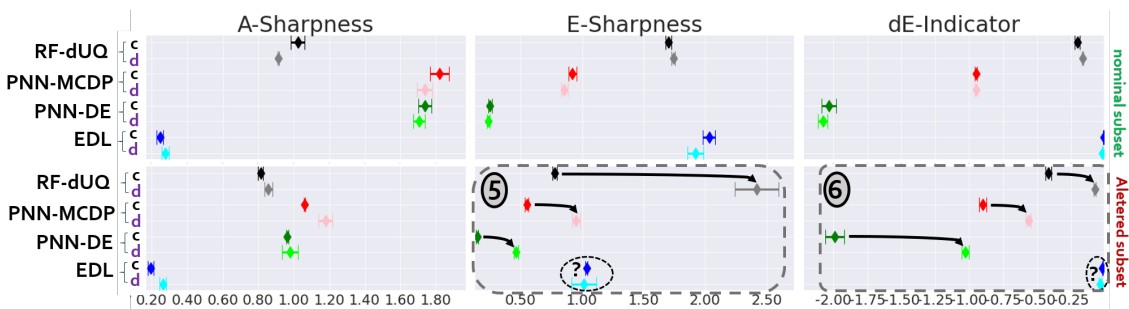

Figure 10: dUQ indicators of the 4 approaches on nominal and altered subset, for a learning variability injection on synthetic data.

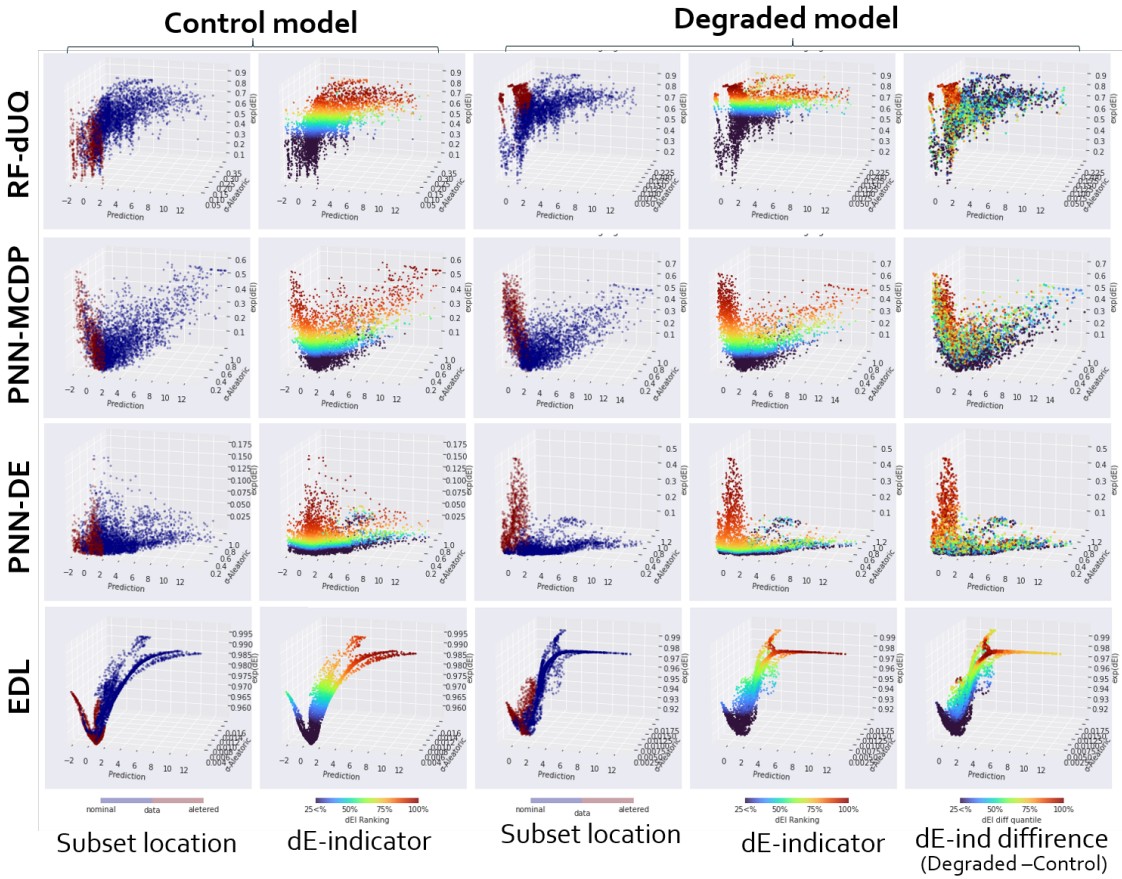

Figure 11: dUQ space for degraded and control versions of the 4 approaches on synthetic dataset, with learning variability injection on the altered subset.

### E.6 Synthesis of combinatorial of experiments :

| Type of variability injection | Dataset | # Experiments | Altered queries types | Injection Strengh | Cross-val |
|---|---|---|---|---|---|
| Training (Withdrawal) | gas demand | 24 | 3: {low-var; mid-var; high-var} | 2: {97%;100%} | 4 |
| | synthetic | 8 | 1: {ctx1} | 2: {97%; 100%} | 4 |
| Inference (Alteration) | gas demand | 16 | 2: {local; global} | 2: {strong;small} | 4 |
| | synthetic | 16 | 2: {local; global} | 2: {strong;small} | 4 |

Table 5: Resume of the various experiments for both variability injection type on both datasets.

## F  Implementation details

We provide all the implementations and datasets via a Github repository : https://github.com/AnonymousSubmission6667/disentangled-Uncertainty-Quantification. The developments are performed using standard ML libraries (Scikit-Learn Pedregosa et al. (2011) and TensorflowAbadi et al. (2016)) and we have used CPU processing unit for all the experiments.

Various notebooks are provided in our Github repository which allow to reproduce the obtained results presented in the paper. They are accessible from the `notebook` directory located at the root of the repository. The details concerning the installation of dependencies are also brought in the `README` file of the Github repository.