# OpenReview forum: "Experimental methodology to evaluate the effectiveness of uncertainty disentanglement on regression models"
_ICLR.cc/2024/Conference — Submitted to ICLR 2024_

### Official Review · Reviewer_wzW6 · 2023-10-30

**Soundness:** 1 poor
**Presentation:** 1 poor
**Contribution:** 2 fair
**Rating:** 1
**Confidence:** 5

**Summary:**

In this work, the authors propose a framework for measuring the effectiveness of modern uncertainty quantification methods (UQ) for the task of disentangled uncertainty quantification (dUQ), which attempts to factorize the uncertainty of a model into reducible epistemic uncertainty and irreducible aleatoric uncertainty. To this end, the authors propose the disentangled Epistemic indicator (dE-Ind), corresponding to a ratio of predicted aleatoric to epistemic uncertainty from their qQU model output, and novel epistemic injection schemes for training and inference. The authors demonstrate their framework across 5 different modern ML UQ models and several datasets.

**Strengths:**

- The authors conduct a thorough analysis over recent methods in the field of uncertainty quantification in application to their problem setup.
- The problem of measuring epistemic uncertainty, with no ground truth, is an important question, and the authors initially positioned their work nicely in context with respect to related work.

**Weaknesses:**

- There is a lack of clarity in the story of this paper, and what the authors consider their main contributions. In the introduction, they state that "In this article, we seek to bridge this gap by introducing a dUQ evaluation methodology providing a comprehensive benchmark of various typologies of UQ models." but later on, in section three, they state that "The experimental goal is to highlight an epistemic confidence gap in model predictions, between nominal and altered queries (i.e. affected by injections of epistemic uncertainty)."
- The writing is very dense and the presentation makes the content more difficult to understand. In particular, Figures 1 and 3 have to much detail and are too difficult to follow in tandem with the text, which is itself not precise and can be quite confusing. Figure 2 is also unclear as to where we should be paying attention, or what insight to draw from it. For an example of a sentence I had difficulty understanding: "These two components are defined for and in a given modeling scope, resulting from the methodological choice of both observed features and model type, which depends on the predictive task’s characteristics".
- The paper has an entire section dedicated to training epistemic injections, but this is never mentioned before as a part of their contribution or utility. The beginning of the paper makes it seem like they offer an evaluation of modern methods, but this along with the meta-model seem to be proposing training procedures?

**Questions:**

- Why is the training set parameterized by theta? (above equation 1 on page 3)
- Is Θ related to all different partitions of a particular dataset, or from the true distribution of data?
- What is meant above Figure 1 with "the previous negligible terms can then induce blurs into the uncertainty decomposition."?
- What is the mentioned variability infusion mechanism referring to?
- When Meta-models are introduced they are not explained. I see that they are mentioned later in the results section, but at a high level are these supposed to be neural networks?
- What was the rational behind the equation for dE-Ind? Why that particular ratio?
- What are queries in this context? The term is introduced at the beginning of section 3, but it is unclear what it represents. Similarly, epistemic query injections are discussed by never defined.
- What was the intention with the inclusion of section 4.1? Was it to evaluate the existing methods on these new benchmarks?
- It's very difficult to parse the analysis for the results in Figure 4. What are the high level takeaways?
- In Table 3, what is meant by "no cross validation due to size of the dataset"?

---

> ### Author Response · Authors · 2023-11-22
> **Answer to review 3**
>
> W1) Indeed, the link is not necessarily clear: starting from the assumption established in Section 4.1 that the 4 models produce a relevant uncertainty quantification, we then use "epistemic confidence gap" due to our epistemic uncertainty injection mechanism as an evaluation tool (dE-ind shift measure) to characterize the "UQ disentanglement relevance". Furthermore, we investigate whether the disentangled epistemic component makes sense.
>
> W3) There may be some confusion due to the use of the term "training epistemic injection", which simply refers to the step where the injection takes place. The training-step injection is a scheme of evaluation that requires alteration of a subpart of the dataset during training. In contrast to adversarial learning, at this time, we do not envisage a (meta)-model learning procedure incorporating epistemic injection mechanisms, even if the idea makes perfect sense and could arise in the close future.
>
> Q1) The notation is ambiguous: The training set is not “parameterized” but is related to theta. Rather, it should be understood as starting from the sub-set of data called $D_\theta$ according to a type of model $\hat{f}$ (with deterministic optimization procedure), we obtain a trained model $\hat{f}_{\theta}$.
>
> Q2) Indeed, $\Theta$ is related to all the different partitions of a particular dataset (which corresponds to the observed data used for model training). Moreover, it's the "meeting" between all different partitions and a type of model, which produces all reachable sets of parameters for this considered model in the scope defined by observed data. It is by analyzing the stability of responses on this set of virtual models (in an approximate way) that we can access a form of epistemic confidence measure.
>
> Q3) Deriving equation 2 is made using several assumptions (strong independence and zero-mean), to neglect terms that complexity the decomposition. When these assumptions cease to be valid, the decomposition into epistemic and aleatoric components becomes less obvious in its mathematical form, and the clear line between the two components is blurred by interaction terms. Some additional intuition is provided in Appendix C3
>
> Q4) The variability infusion mechanism mentioned on page 4 is a generic term referring to the fact that uncertainty quantification requires a diversity of predictions to be performed. To do so, there exist several possible mechanisms aiming at "simulating" the variance of model predictions. The choice of mechanism depends on the UQ paradigm and this variability is at the heart of epistemic uncertainty quantification.
>
> Q5) On page 2, we state that a metamodel is a way to synthesize an ensemble of outputs produced by an ensemble of submodels. Thus, metamodels are not necessarily neural networks. Rather, "metamodel" is a generic term encompassing several ways to compute an epistemic confidence level by aggregating submodels outputs (note that submodels can be neural networks).
>
> Q6) A query is when we ask a model to make a prediction on a particular instance. We do not use the terms "epistemic query injections" together in the paper but "epistemic injections" refers to mechanisms aiming at increasing epistemic uncertainty (by “injecting” variability) and an "altered query" is when we ask a model to predict a OOD instance, made by epistemic injection.
>
> Q7) Section 4.0 aims to illustrate that our implementations of the 4 approaches are competitive in terms of UQ (total variance) on public datasets. Section 4.1 aims to illustrate that the UQ (total variance) performances of our 4 approaches are comparable to our study datasets. This allows us to move on to the next step: the approaches also claim to produce uncertainty disentanglement, so how to evaluate it? Here, they were assessed under the prism of the epistemic measure reaction to simulated OOD data.
>
> Q8) Thank you for the feedback. This figure has to be reworked, its main lessons are :
> - a) Proof that experimental setup is coherent:  For an epistemic injection linked to an altered subset, we clearly show that the performances and UQ indicators values (Aleatoric and Epistemics) are close to the non-altered subset for both control and degraded models. In contrast, we observe an impact (performance loss) between the control and degraded model for the altered subset.
> - b) Proof that we can claim a dUQ effectiveness : The degraded model expresses a more significant rise (shift of confidence) of UQ epistemic measure than its aleatoric counterpart. The dE-indicator synthesizes both of these observations in one value.
> - c) Unlike the three other approaches, EDL does not show this expected behavior, which questions the effectiveness of its uncertainty decomposition.
>
> Q9) For the year prediction dataset, which is much larger than other public dataset, we did not reproduce the experiments four times for each of our four approaches, so there is no measure of performance deviation.

---

### Official Review · Reviewer_k1SL · 2023-10-31

**Soundness:** 1 poor
**Presentation:** 1 poor
**Contribution:** 1 poor
**Rating:** 1
**Confidence:** 4

**Summary:**

This paper considers the problem of evaluating epistemic uncertainties outputted by a Gaussian auto-regressive model. The authors proposed an injection scheme where the model is asked to make predictions on modified inputs, and the epistemic uncertainty can then be compared between the raw input and modified inputs.

**Strengths:**

This paper examines an important question of evaluating epistemic uncertainty, which remains challenging in predictive ML. On the top level, modifying inputs and checking how the predicted uncertainties vary sounds intuitively promising and could lead to interesting future developments.

**Weaknesses:**

However, on the negative side, this paper has critical issues with the soundingness of the approach and the overall clarity.

For the soundingness of the approach, one fundamental problem in this paper was that the proposed (dE-Ind) didn't get any theoretical justification. While intuitively, the change in the proportion of epistemic uncertainty might be an interesting metric to check, it doesn't solve the essential problem of there isn't a so-called ground truth with epistemic uncertainty. In fact, there has been recent work that indicates there might not be a proper loss for the epistemic uncertainty that can be evaluated with data (https://arxiv.org/abs/2301.12736). Furthermore, the injection schemes to compute the proposed metric also lack support. There isn't a clear discussion on how the proposed injection was the correct one to generate inputs that provide the suitable change of uncertainty. The description of methods is also quite handwavy, as some steps are just given without any background (i.e. use SHAP to modify input features).

On the clarity part, this paper doesn't have consistent mathematical notations and might cause some significant issues for the readers. The boldface x is used both with the time index as part of a time series and with the instance index as a training example. The function notation f and expectation notation E use the bold and non-bold font without any apparent difference. Additional marks like \bar, \hat, * are all directly given without a clear introduction.

**Questions:**

I wonder if the authors can give any insight on how the injection method, together with the dE-Ind metric, could give the correct evaluation of epistemic uncertainty, and what is the optimal epistemic uncertainty with your defined metric?

---

> ### Author Response · Authors · 2023-11-22
> **Answer to review 2**
>
> W1) The dE-indicator is used as an evaluation tool as part of our evaluation methodology. It corresponds to a normalization of the epistemic indicator by the total variance. Its incentive is to make it easier to observe the epistemic shift by considering the heteroscedasticity present in the data.  It also aims to check that it is indeed in the epistemic component and not in the aleatoric one that we find the greatest impact linked to OOD queries. Theoretical justifications are provided in the appendices (in particular Appendix C.5.) based on strong assumptions about the UQ-aleatoric and UQ-epistemic modeling scope. If these justifications are insufficient or unclear, we are open to feedback and criticism.
>
> W2) We may have been ambiguous about the purpose of the paper. Some elements of context may be missing and have been lost while reducing the number of pages to fit publication requirements. We do not pretend to solve the epistemic uncertainty evaluation problem. We even consider that the epistemic uncertainty evaluation is not solvable as one block, and that it will be necessary to set up a complex evaluation/certification protocol aimed at evaluating different aspects and requiring expert knowledge to guide and limit the evaluation perimeter. In this case, on our 2 datasets, and for the types of OOD corresponding to our epistemic uncertainty injection mechanisms (simple, or naive), we demonstrated a robust statistical framework to assess the relevance of the aleatoric vs epistemic decomposition of the 4 approaches studied. This does not cover all possible forms of OOD data, and the methodology has a clear limitation: the injection must be designed in relation to the form of OOD data we are trying to validate, which in practice is often unknown.
>
> W3) First, we simply compare four dUQ approaches from the literature, that we have reimplemented. Our article does not propose a new approach or a new loss based on our dE-Indicator. The intuitions we shared seem to be in accordance with the article you shared: second-order UQ approaches only estimate the aleatoric part, it is necessary to have a meta-modeling mechanism (on top of optimization) to produce an epistemic measure. Our contribution aim is to produce measures enabling us to compare the effectiveness of UQ disentanglement of approaches having different meta-modeling mechanisms.
>
> W4) Epistemic uncertainty injection schemes aim to produce "epistemic noise" leading to a loss of confidence (i.e. lower values of epistemic-confidence indicators) of the model. We don't think there is a "correct OOD input": OOD inputs can take various forms depending on the application. But for a given distribution, we can produce OOD inputs of various complexity and refinement.
>
> A model that claims to perform epistemic uncertainty quantification has to be sensitive to such OOD inputs: we think this is the minimum-expected behavior. To illustrate this point, we designed two kinds of epistemic injection:
>
> -a) The naivest one consists in disrupting feature values to produce explicit OOD (inputs whose values do not follow the expected distribution). However, how can we ensure that this perturbation has a real impact on a regression model? By perturbing features with high contributions to the model decision. In this case, three methods (Mean decrease in impurity for RF-features importance, along with Shap and Sage libraries) produced a ranking of feature contributions.
> -b) As this first type of injection was very naive and produced coarse outliers, we designed another one slightly more sophisticated and producing more convincing OOD : by removing (using expert knowledge of the data) a contextual subset of the training dataset. It is possible to design many other types of epistemic uncertainty injection, producing different types of OOD data. Our paper focuses on the following: does this evaluation methodology based on epistemic injections provide insights for characterizing the ability of a model to express its lack of epistemic confidence?
>
> W5) Thank you for your comments. We're going to track down these errors to clarify the notations. We are reworking the notations to harmonize them and introduce them more effectively. In this case, we use the hat to identify estimates, and bar to identify averages, which may be either average of quantities or average of estimates.
>
> Q1) Perhaps we were ambiguous but we do not claim "correct evaluation of epistemic uncertainty". We are just presenting some exploratory work that has focused on the analysis of a measure (dE-Ind) which can be interpreted as a normalized epistemic confidence measure. This normalization is realized to better handle data heteroscedasticity. This analysis aimed at characterizing part of the epistemic evaluation problem: namely, the ability to process the OOD inputs produced by our epistemic injection mechanisms.

---

### Official Review · Reviewer_3UD4 · 2023-10-31

**Soundness:** 3 good
**Presentation:** 2 fair
**Contribution:** 2 fair
**Rating:** 3
**Confidence:** 4

**Summary:**

The authors propose a new framework for evaluating different approaches for UQ in regression models.  They propose two types of epistemic variability injections in order to quantify a confidence gap of a model.

**Strengths:**

The authors address the important task on how to quantify the qulaity of UQ methods; I appreciate the introduction of 2 new dataset, a real one and a synthetic one, which helps towards a more realistic evaluation

**Weaknesses:**

- The authors missed a while section of the literature,  on calibration for regression models, which covers a highly related task. In particular [1] demonstrates how calibration quantifies a meaningful tradeoff between sharpness and coverage
- I found the concepts of Inference step injection of epistemic variability injections and Training step injectionare fairly straight-forward concepts, which are presented in a convoluted manner - here the clarity of the paper could be improved substantially
- It's not clear how the choices of the type of "attack" in the inference step rejection influence the conclusion - surely the way outliers are sampled from the tail (how far) is relevant? Similarly, how altered is the altered dataset and hwo does this effect conclusions? To me the main contribution is the benchmark study and this would need a much more exhaustive and systematic set of experiments


[1] Kuleshov, V., Fenner, N., & Ermon,  S. (2018). Accurate uncertainties for deep learning using  calibrated regression. In International conference on machine learning (pp. 2796-2804). PMLR.

**Questions:**

See above

---

> ### Author Response · Authors · 2023-11-22
> **Answer to review 1**
>
> W1) Thank you for the reference. We are familiar with the calibration curves and should have mentioned them along with the assessment of total uncertainty. We computed curves of this kind, but we considered that they didn't add any value here, as the question of assessing total uncertainty is secondary (our paper focuses on the epistemic part). However, we disagree in part. Calibration curves, which provide more information than simple X\% coverage, cannot totally replace a sharpness measure: the paper seems to mention that part  "3.1 Paragraph Calibration and Sharpness".
>
> W2) Thank you for this feedback, we'll be thinking about how to clarify this section. The idea behind the type of attack is to modulate the nature of the OOD queries created to see if different behaviors can be observed in the epistemic response of the models. Rather than simply playing on the “magnitude” (= how far), we proposed other kinds of context-conditional OOD.
>
> Theoretically, experiments with epistemic injections designed to produce stronger OOD queries (=inputs) would be linked to stronger test statistics due to a stronger shift in epistemic confidence between the control model and the degraded model. This phenomenon is observed in several cases for specific configurations. for example, we always observe higher test values for the same experiment of inference step injection, by modifying only magnitude from small vs strong.
>
> However, we did not observe such a trend when studying the effect of the different context-conditional OOD  (for example between local and global). It could be explained because models do not necessarily apprehend OOD data as we might think, since it depends directly on their modeling mechanisms. For example, let's consider an RF model, and an OOD instance located a little further away than the data domain border: if we produce a new OOD instance by doubling the values of this instance, both instances will be perceived as identical (same partitions) for the model in terms of epistemic confidence, although the latter can be considered a far OOD than the former.
>
> W3) It is difficult and time-consuming to exploit and interpret the results one by one. But, the aggregation of results nevertheless provides good insight for the comparison between approaches, as it illustrates that the models do not have the same sensitivity to these different kinds of OOD queries. Based on your feedback, we will think about a modification in the choice and presentation of the combinatorial experiments linked to the injection parameters.

---

### Author Response · Authors · 2023-11-22
**General answer**

Firstly, We apologize for the late response to your feedback. We would like to thank the reviewers for taking the time to read and comment our paper in spite of its density, resulting from trying to fit numerous details of our methodology in a limited number of pages. There may indeed be some ambiguity about the intentions and goals of this paper. We take note that it is necessary to rework the form of this paper to make it less dense and clearer for a future submission.

To clarify some ambiguity: we do not pretend to solve the epistemic uncertainty evaluation problem. We even consider that the epistemic uncertainty evaluation is not solvable per se and that it will be mandatory to set up a complex evaluation/certification protocol aimed at evaluating different aspects of this issue and requiring expert knowledge to guide and limit the evaluation perimeter.

We only aimed to provide an experimental methodology based on epistemic uncertainty injection (that we have applied to our 2 datasets). In doing so, we wanted to produce insights about how some dUQ models (that provide epistemic confidence measures) react to OOD input, created with our epistemic uncertainty injection mechanism.

So, on our 2 datasets, and for the types of OOD corresponding to our different epistemic uncertainty injection mechanisms (simple, or naive), we implemented a robust statistical framework giving insights about the relevance of the aleatoric vs epistemic decomposition of the 4 approaches studied. This does not cover all possible forms of OOD data, and the methodology displays a clear limitation: the injection must be designed for a specific form of OOD data to validate, which in practice is often unknown.

---

### Meta-Review · Area_Chair_Eg9q · 2023-12-05

**Metareview:**

This paper explores the evaluation of uncertainty quantification for regression models.  The review scores were 1, 1, 3 putting the paper unfortunately well below the bar for acceptance.  The reviewers seemed to have considerable trouble understanding the contribution from the authors and overall didn't seem to think the paper was ready for acceptance. Some common concerns among the reviews included missing discussion of relevant literature, disorganized presentation and lack of clarity.  Hopefully the reviews will be helpful to the authors to strengthen the manuscript for a future submission.

**Justification For Why Not Higher Score:**

Review scores were very, very low.

**Justification For Why Not Lower Score:**

Almost impossible.

---

### Decision · Program_Chairs · 2024-01-16

Reject